# Biosynthesized Silver Nanoparticles for Cancer Therapy and In Vivo Bioimaging

**DOI:** 10.3390/cancers13236114

**Published:** 2021-12-04

**Authors:** Shagufta Haque, Caroline Celine Norbert, Rajarshi Acharyya, Sudip Mukherjee, Muralidharan Kathirvel, Chitta Ranjan Patra

**Affiliations:** 1Department of Applied Biology, CSIR-Indian Institute of Chemical Technology, Uppal Road, Tarnaka, Hyderabad 500007, Telangana, India; shagufta.220@csiriict.in (S.H.); carolinenorb@gmail.com (C.C.N.); acharyyarajarshi@gmail.com (R.A.); sudip.mukherjee1988@gmail.com (S.M.); muralidharan@csiriict.in (M.K.); 2Academy of Scientific and Innovative Research (AcSIR), Ghaziabad 201002, Uttar Pradesh, India

**Keywords:** silver nanoparticles, green chemistry approach, *Zinnia elegans*, cancer therapeutic, *NIR* bioimaging, nanomedicine

## Abstract

**Simple Summary:**

Chemotherapy, a conventional treatment strategy, is associated with several limitations. Alternatively, cancer nanotechnology offers new strategies for remedy from drug delivery and extending to therapeutics using nanoformulation. Recently, silver nanoparticles were utilized for different cancer theranostics. In this context, the current manuscript demonstrates the design and development of biologically synthesized silver nanoparticles (AgZE) and their applications for cancer theranostics. The synthesized AgZE is optimized and thoroughly characterized. The AgZE exhibits biocompatibility and selective anticancer activity towards the cancer cell lines established through various assays. The fluorescence properties of AgZE are observed in the *NIR* region (excitation: 710 nm, emission: 820 nm) in the normal and tumor bearing C57BL6/J mice. The silver of AgZE is found to be biodistributed in different vital organs as analyzed by ICPOES. Thereafter, these results highlight that AgZE could be an efficient cancer therapeutic and *NIR* based non-invasive imaging agent in the upcoming times.

**Abstract:**

In the current communication, a simple, environmentally compatible, non-toxic green chemistry process is used for the development of silver nanoparticles (AgZE) by the reaction between silver nitrate (AgNO_3_) and the ethanolic leaf extract of *Zinnia elegans* (ZE). The optimization of AgZE is carried out using a series of experiments. Various physico-chemical techniques are utilized to characterize the nanomaterials. The cell viability assay of AgZE in normal cells (CHO, HEK-293T, EA.hy926, and H9c2) shows their biocompatible nature, which is supported by hemolytic assay using mouse RBC. Interestingly, the nanoparticles exhibited cytotoxicity towards different cancer cell lines (U-87, MCF-7, HeLa, PANC-1 and B16F10). The detailed anticancer activity of AgZE on human glioblastoma cell line (U-87) is exhibited through various in vitro assays. In vivo the AgZE illustrates anticancer activity by inhibiting blood vessel formation through CAM assay. Furthermore, the AgZE nanoparticles when intraperitoneally injected in C57BL6/J mice (with and without tumor) exhibit fluorescence properties in the *NIR* region (excitation: 710 nm, emission: 820 nm) evidenced by bioimaging studies. The AgZE biodistribution through ICPOES analysis illustrates the presence of silver in different vital organs. Considering all the results, AgZE could be useful as a potential cancer therapeutic agent, as well as an *NIR* based non-invasive imaging tool in near future.

## 1. Introduction

Cancer is considered the foremost basis of mortality as well as morbidity worldwide. Millions are being affected by cancer every year as per global statistics, with 19.3 million new cases, followed by deaths of 10 million, in the year 2020 [1,2,3,4]. The conventional treatment therapies available for cancer treatment are surgery, chemotherapy, radiation therapy, hormonal therapy, and immunotherapy, with chemotherapy being the most popular. However, all of these conventional therapies have several limitations, including bioavailability, non-specificity, and toxicity [5,6]. Considering these limitations, researchers in the area of biomedical sciences have revolutionized cancer therapeutics with nanotechnology [7,8,9,10]. Nanotechnology is explored for cancer therapeutics and diagnostics using different nanoparticles as they aid either through active (using targeting agent) or passive (by enhanced permeability and retention effect) mechanisms for increased efficiency and low toxicity [7,8,9,10,11]. Among the different inorganic nanoparticles, silver is the most popular nanoparticle for cancer therapeutics [7,8,9,10,12]. However, the silver nanoparticles produced from the chemical and physical methods have certain disadvantages, which mainly include the use of harsh environmentally degrading chemical reagents and reaction protocols that use a lot of energy producing considerable amount of waste. Henceforth, scientists are focusing on the silver nanoparticle synthesis using biological sources such plant and animal extracts [8]. Biosynthesis is a feasible, easy, ecofriendly, and non-toxic method. Our group has already developed several biosynthesized nanoparticles (silver- and gold-based) using different sources of plant leaf extract, and demonstrated their biological properties in various ways. For example, biosynthesized silver nanoparticles using *Olax scandens* shows the potential application as drug delivery system along with fluorescence properties and anticancer, as well as antibacterial, activity [13]. Gold nanoparticles synthesized using *Eclipta alba* show biocompatibility and would be useful for drug delivery applications [14]. On the other hand, gold nanoparticles synthesized using *Hamelia patens* extract show pro-angiogenic properties [15]. Therefore, different nanoparticles produced with different plant extracts show varying biological properties due to the presence of various active phytochemicals and reducing agents. The biosynthesis of nanoparticles is a bottom-up process with biological agents assisting reduction, capping, and stabilization [13,16,17,18].

Other than the anticancer therapeutics, nanotechnology is also explored for disease diagnosis using their unique physico-chemical features such as radioactivity, fluorescence, magnetic properties, high electron density in nature, etc. [19]. Recently, non-invasive *NIR*-region (700–900 nm) based fluorescence imaging has been produced using nanotechnology. The *NIR* photons dig deeper into the biological tissues as compared to visible light, providing information about their structural and functional features without damage [20,21,22]. Different chemical based-*NIR* fluorophores are available but are associated with several limitations such as low sensitivity, issues with tissue penetration, large photon scattering losses, toxic in nature, etc., resulting in many opting for nanotechnology [6,19,23,24,25,26,27]. Recently, using nanotechnology, the *NIR* based fluorescent nanoparticles, especially those that are biologically synthesized, are preferred for non-invasive bioimaging [28]. To this end, our group already reported biosynthesized gold nanoparticles (AuZE), obtained by the reduction of gold salts using extracts of *Zinnia elegans*, that exhibit fluorescence at both the green (emission: 450 nm; excitation: 350 nm) and red regions (emission: 720 nm; excitation: 450 nm). The AuZE when injected intraperitoneally in C57BL6/J mice exhibited non-invasive *NIR* bioimaging (excitation: 710 nm; emission: 820 nm) [29]. However, these gold nanoparticles do not exhibit anticancer activity unless we use any chemotherapeutic drug. Hence, design of *NIR* based fluorescent nanoparticles having its own anticancer activity (without any drug) is always preferable for the study of cancer theranostics. Considering the above fundamental issues and knowing the anticancer activity of silver from various published literature, we designed and developed silver nanoparticles (AgZE) using the leaf extract of *Zinnia elegans* (ZE). The *Zinnia elegans* plant has been selected for biosynthesis as it emits fluorescence in the *NIR* region as per the published literature [29]. Additionally, the ZE plant has several medicinal values such as hepatoprotective, antifungal, antihelmintic, antimalarial, anti-infective, phytoremediation, etc. [29,30,31]. Our hypothesis is that, during synthesis, some of the bioactive molecules can attach to the surface of the nanomaterials that show some biological activity. Interestingly, the biocompatible AgZE exhibits efficient anticancer activity as well as in vivo *NIR* based imaging (excitation: 710 nm; emission: 820 nm). The anticancer properties of the AgZE are thoroughly studied in vitro with its underlying mechanisms. The non-invasive in vivo imaging of AgZE is evaluated in the C57BL6/J mice (with and without tumor) illustrating their potential bioimaging activities. These new observations of anticancer as well as *NIR*-based in vivo imaging indicates AgZE as a theranostic agent with potential applications in future research.

## 2. Materials and Methods

### 2.1. Materials

Silver nitrate (AgNO_3_), ethanol (EtOH), MTT, ribonuclease (RNase), trypsin, PI: propidium iodide, FBS: fetal bovine serum, DMEM: Dulbecco’s modified Eagle’s medium, penicillin, protease inhibitor cocktail (PIC), RIPA buffer, DCFDA: 5-(and-6)-carboxy-2′,7′-dichlorofluorescein diacetate, glycine, acrylamide, tris-base, ammonium persulfate, crystal violet, and PBS: phosphate-buffered saline are procured from Sigma Aldrich Chemicals, St. Louis, MO, USA. Methanol (MeOH) is procured from Merck Specialties (India). Tween-20 is procured from Amresco (USA). Triton X-100 and Bradford reagent are brought from Genetix Biotech Asia. Dimethyl sulphoxide (DMSO) is purchased from Rankem (India). Glycine, acrylamide, bis-acrylamide, and sodium bicarbonate (NaHCO3) are purchased from HiMedia (India). PVDF membrane is procured from Merck Milipore. All the experiments have been carried out in Milli-Q-grade water (18.2 MΩ.cm).

Primary antibodies anti-p53 mouse mAb, anti-caspase-8 mouse mAb, anti-GAPDH rabbit mAb, anti-STAT3 rabbit mAb antibodies as well as secondary goat anti-mouse IgG and anti-rabbit, HRP-linked antibodies are procured from Cell Signalling Technologies, Danvers, MA, USA. Protein ladder and chemiluminescent substrate are purchased from Thermo Fischer Scientific, Waltham, MA, USA.

The plant leaves of *Zinnia elegans* have been collected from CSIR-IICT garden as per our previously published literature [29].

#### 2.1.1. Cell Lines

The cell lines procured from NCCS, Pune, India are (i) MCF-7: human breast cancer cell line, (ii) PANC-1: human pancreatic cancer cell line, (iii) HeLa: human cervical cancer cell line and iv) B16F10: murine melanoma cell line. The cell lines purchased from ATCC, USA are (i) U-87: human primary glioblastoma cell line, (ii) CHO: Chinese hamster ovary cell line, (iii) HEK-293T: human embryonic kidney cell line, and (iv) H9c2: rat cardiomyoblast cell line.

#### 2.1.2. Preparation of Ethanolic *Zinnia elegans* Plant Leaf Extract

The freshly collected leaves of *Zinnia elegans* (ZE) are thoroughly washed with water, dried, weighed, and grinded in ethanol for extract preparation. The extract is then centrifuged at 10,000 rpm for about 30 min under ambient temperature. The supernatant is collected and stored in −20 °C as per our standard laboratory procedure in published literature [29].

#### 2.1.3. Synthesis of Silver Nanoparticle Using Ethanolic ZE Plant Leaf Extract

The protocol for biosynthesis of silver nanoparticles (AgZE) follows our previously published literature [13,29]. Briefly, 250 µL of 10^−2^ (M) silver nitrate is added to the beaker containing water followed by addition of diluted ZE extract (100–500 µL) dropwise (total volume of the solution is 5 mL) and allowed to stir for 72 h. The reaction is optimized using different reaction sets with varying concentrations of ZE extract (100–500 µL; 1:1 dilution in ethanol) keeping the concentration of silver nitrate (250 µL of 10^−2^ M) constant (Appendix A). The UV visible spectroscopy reading is taken after completion of the reaction to check the wavelength of the light absorbed by the silver nanoparticles (AgZE) using Synergy H1 microplate reader.

### 2.2. Characterization

The characterization of AgZE is performed by different analytical techniques (DLS, XRD, FTIR, SEM, TEM, fluorescence, and ICPOES) following earlier published literature [29]. The characterization techniques are discussed in Appendix A.

### 2.3. In Vitro Experiments

#### 2.3.1. Cell Culture Experiments

The cancer cell lines (U-87, PANC-1, MCF-7, HeLa, and B16F10) and normal cell lines (HEK-293T, CHO, H9c2, and EA.hy926) are cultured in DMEM media supplemented with 10% FBS, 5% L-glutamine, and 1% antibiotics (penicillin-streptomycin) in a humidified 5% CO2 incubator at 37 °C for all in vitro experiments [32].

#### 2.3.2. Cell Viability Assay Using MTT Reagent

As per well-established literature, in this present manuscript, the normal cells (CHO, HEK-293T, EA.hy926, and H9c2) are incubated for 24 h and 48 h with AgZE, whereas cancer cells (HeLa, B16F10, MCF-7, PANC-1, and U-87) are incubated for 48 h before cell viability assay using MTT reagents [7,10,11,33]. MTT is employed to test cell viability of AgZE on the normal (CHO, HEK-293T, EA.hy926, and H9c2) as well as on the cancer cell (MCF-7, U-87, PANC1, HeLa, and B16F10) lines according to our published literature [7,10,11,33]. Briefly, the normal and cancer cells are seeded into 96-well plate (8000 cells/well), incubated, and treated with AgZE in concentration dependent manner (details in Appendix A).

#### 2.3.3. ICPOES for Determination of Silver

The silver content in (i) AgZE pellet, (ii) cells, and (iii) tissues of vital organs is determined using ICPOES according to our earlier published protocols [29,32].

Initially, the silver content in an AgZE pellet is analyzed by ICPOES analysis, where 100 µL of the pellet is suspended in 1 mL of 70% nitric acid, incubated for 2 h, filtered, and the final volume is made up to 50 mL using MilliQ water and submitted for ICPOES analysis using IRIS intrepid II XDL, ThermoJarrel Ash. The experiments are carried out in triplicate [33].

The cell uptake analysis of the AgZE in normal (HEK-293T) and cancer (U-87) cell line is studied using the same ICPOES analysis. Both the cells are plated in the T-25 flasks (density 15 × 10^5^ cells/plate), allowed to grow till 80% confluency and treated with AgZE for 24 h. Later, the cells are washed with DPBS (3×) for removing unattached particles followed by trypsinization, counting, digestion with 70% HNO_3_, and filtering, and the final volume is made up to 50 mL for ICPOES analysis of silver.

Similarly, the vital organs (liver, brain, kidney, spleen, heart, lung, and colon) are collected after sacrificing the mice, weighed, and digested in 70% nitric acid and submitted for ICPOES analysis to detect silver content in AgZE.

#### 2.3.4. Scratch Assay

The scratch assay using AgZE is carried out to check its anti-migration effect on U-87 cancer cells as per previous published literature [32]. Briefly, U-87 cells (5 × 10^4^/well) in 24-well plate are seeded for 80% confluence. An in vitro wound model is generated by scratching on monolayer of cells using pipette tip (sterile), washed with 1× PBS and treated with AgZE. Temozolomide (TEMO) is used as positive control (anticancer drug). The bright field images of scratched area of cells are taken at 0 h and 7 h time intervals under an inverted microscope (Nikon, Tokyo, Japan) at 4× magnification. The wound closure is calculated with the help of IMAGE J software analysis (NIH, Bethesda, MD, USA).

#### 2.3.5. Transwell Migration Assay

The transwell based cell migration assay is performed as per previously published literature [34,35]. Initially, 4 × 10^4^ cells are suspended in 200 μL of DMEM medium with respective treatments (i) untreated, (ii) TEMO, and (iii) AgZE (IC50: 4.1 μL = 0.94 μg/mL). The cells are seeded into the upper chamber of separate inserts placed in 24-well plate. Next, 500 μL of DMEM is added to the 24-well plates. After incubation at 37 °C for 14 h, the migrated cells are washed with PBS, then fixed with 4% paraformaldehyde, washed with PBS, permeabilized with 100% methanol, stained with 0.1% crystal violet solution in PBS. Finally, cells are again washed with PBS and dried. The cells in the upper side of inserts are carefully removed using swabs and observed under an inverted microscope (Nikon, Tokyo, Japan) at 4× magnification.

#### 2.3.6. Cell Cycle Assay

The effect of AgZE on cell cycle progression of cancer cell line (U-87) is checked using propidium iodide staining as per published literature (details in Appendix A) [32].

#### 2.3.7. Cellular Apoptosis Assay

The apoptotic effect of AgZE on U-87 cancer cells is analyzed by cell apoptosis assay using Annexin-V as well as propidium iodide stain in flow cytometry (details in Appendix A) [32].

#### 2.3.8. Reactive Oxygen Species (ROS) Assay

ROS generation ability of AgZE on U-87 cells is carried out as per earlier published protocol (details in Appendix A) [36,37].

#### 2.3.9. Western Blot Analysis

The Western blot analysis for anticancer activity of AgZE on U-87 cells (incubation or stimulation time 18 h) is evaluated according to a previously published protocol (details in Appendix A) [33]. All the whole western blot figures can be found in the Appendix A.
% of Normalization: values of test protein/values of loading control protein.

#### 2.3.10. Cellular Imaging by Confocal Microscopy

The cellular uptake of AgZE is analyzed by laser scanning confocal microscopy (Nikon). Initially, CHO (non-cancerous), U-87, and B16F10 (cancer) cells on sterile cover slips and incubated with AgZE as per an earlier published protocol [29]. After 6 h of treatment, the cells are thoroughly washed with DPBS and fixed in 4% paraformaldehyde solution. The fixed cells are mounted using Fluoroshield DAPI media (Sigma Aldrich) on glass slides. The slides are observed under a confocal microscope using DAPI and LDS 751 channels. The images are processed using the NIS element imaging software.

### 2.4. In Vivo Assays

#### 2.4.1. Chorioallantoic Membrane (CAM) Assay

Chorioallantoic membrane (CAM) assay is used to investigate the effect of AgZE during the developmental stages of chick on blood vessels of chorioallantoic membrane as per published report (details in Appendix A) [29].

#### 2.4.2. Non-Invasive Imaging In Vivo

The in vivo experiments are conducted as per the protocol approved by the Institute of Animal Ethics Committee (IAEC) at CSIR-IICT, Hyderabad; vide approval # IICT-IAEC-17-2021 dated 20 January 2021. The fluorescence feature of the ZE extract, as well as AgZE, is evaluated for in vivo imaging in the mice (C57BL6/J) as per published literature [29]. The mice, intraperitoneally injected with the respective treatments, are divided into three groups, namely (i) untreated, (ii) ZE extract (200 μL of the ethanolic solution lyophilized and dispersed in water), and (iii) AgZE (200 μL of the pellet) (*n* = 3 for each group and time point). The mice are anesthetized using ketamine plus xylazine cocktail injection for capturing fluorescence images with in vivo imaging system (Perkin Elmer-IVIS Spectrum) at excitation: 710 nm and emission: 820 nm (time points: 4 h and 24 h). For viewing the biodistribution of ZE and AgZE, the mice are aligned in both dorsal as well as ventral position. The mice are then sacrificed after 4 h and 24 h, keeping the vital organs on a glass Petri dish for ex vivo imaging. The untreated animal organs images are captured at similar time points.

#### 2.4.3. Biodistribution Study in Tumor Model

In vivo melanoma tumor model is developed in female C57BL6/J mice by injecting B16F10 cells (~5 × 10^5^ cells resuspended in 100 µL of sterile HBSS buffer) subcutaneously to the lower right abdomen as per earlier published protocol [32]. After ~12 days of tumor inoculation, the mice are intraperitoneally injected with AgZE −200 μL of the pellet. The mice are anesthetized using ketamine plus xylazine cocktail injection for capturing fluorescence images with in vivo imaging system (Perkin Elmer-IVIS Spectrum, USA) at excitation: 710 nm and emission: 820 nm (time points: 4 h and 24 h). For viewing the biodistribution of ZE and AgZE, the mice are aligned in both dorsal as well as ventral position. The mice are then sacrificed after 24 h, keeping the vital organs on a glass Petri dish for ex vivo imaging.

#### 2.4.4. Hemolysis Assay

The hemolysis assay of AgZE is evaluated on mice RBC as per our previous published literature [32]. Initially, 2 mL of mice blood (C57BL6/J) is collected in 5% EDTA coated tubes, centrifuged at 3000 rpm for 10 min at 4 °C using a Labocene centrifuge. The RBC pellet is then washed thrice in DPBS buffer and suspended in 13 mL of DPBS to form the stock of erythrocyte. From this stock solution, for each experiment, 0.1 mL of the erythrocyte sample suspension is added individually to 0.9 mL (i) positive control: tap water, (ii) negative control: DPBS buffer, (iii) AgZE (10, 50, and 100 μL of AgZE pellet is dissolved in DPBS to make it 1 mL), and (iv) ZE (100 μL in 900 μL DPBS). All the experiments are carried out in triplicate. All samples are incubated in water bath shaker for a time period of 90 min at 37 °C, and then centrifuged at 3000 rpm for 10 min at 4 °C. The absorbance of each centrifugate is measured at 514 nm using the Synergy H1 multimode reader by collecting the supernatant.
% of hemolysis = ((O.D. of treatments − O.D. of negative control)/(O.D. of positive control − O.D. of negative control)) × 100

#### 2.4.5. Statistical Analysis

All experiments are performed in triplicate and expressed as the mean ± standard deviation (SD) as well as the mean ± standard error mean (SEM) as applicable. The comparison between two individual groups and multiple groups is calculated using Student’s unpaired *t*-test. * *p* < 0.05 is considered as statistically significant for all experiments.

## 3. Results

### 3.1. Synthesis and Characterization of AgZE

The biological synthesis of the silver nanoparticles is considered to be an ideal approach as compared to the existing methodologies due to use of a green, renewable reagent and a simple, single step reaction protocol [13]. The plant extracts help in the reduction of AgNO_3_ into silver nanoparticles as previously reported by our group [13,38]. In this present study, silver nanoparticles (AgZE) are synthesized using ethanolic (ZE) leaf extract where the extract acts as a reducing, capping agent as well as a stabilizing agent. The ZE extract contains different phytochemicals, polyphenolic/alcoholic compounds, and aldehydes/ketones that help in the reduction of Ag^+^ (in AgNO_3_) to Ag^0^ [8,29,31].

The optimization of the reaction is carried out using a series of experiments by keeping the volume of AgNO3 constant (250 µL: 10^−2^ M) with varying volumes of ZE plant extract (ZE-100, -200, -300, and -500) and labeled as different reaction sets (Appendix A shows representative images) (AgZE-100, -200, -300, and -500; **entry no**: 1–4, respectively) as seen in Appendix A with **entry no. 4** being the optimized set. The appearance of a greenish-yellow reaction mixture indicates (inset of Figure 1a) the formation of silver nanoparticles which is further validated by UV visible spectroscopy. The higher the concentration of ZE extract (in terms of volume), the higher the formation of AgZE nanoparticles is, observed by the intensity of color (Appendix A).

### 3.2. UV-Visible Spectroscopy

The UV absorbance for all the reaction sets of the nanoparticles has been measured in a time dependent manner as provided in Figure 1a and Appendix A. The formation of optimized AgZE (with ZE-500) **entry no. 4** is established by the characteristic UV absorption peak of silver obtained at around 410–420 nm (Figure 1a) from AgZE pellet, corroborating with previously published reports [13,33]. The UV absorbance of **entry no. 4** is higher, illustrating the increased production of AgZE (Appendix A) and the increase in intensity of absorption peak at 410 nm with time depicts the gradual formation of AgZE till 72 h. The AgZE from reaction **entry no. 4** is considered as optimized nanoparticles and used for all characterization techniques, in in vitro as well as in vivo studies.

### 3.3. Dynamic Light Scattering (DLS) Analysis

Dynamic light scattering (DLS) method is an analytical technique that is employed to measure the hydrodynamic diameter and charge of the AgZE nanoparticles that are presented in Figure 1b,c, respectively. Appendix A depicts the size variation of the different sets of AgZE reaction, with **entry no. 4** being the optimized set. The hydrodynamic diameter of the optimized AgZE is found to be nearly 95 nm (Figure 1b). The size chart depicts a small peak (around 950–1000 nm) due to small population of silver nanoparticles with more size and some unreacted plant products within the solution. The various sizes of the nanoparticles obtained by DLS analysis may be due to the presence of strong and weak reducing agents present in the plant extract. It has been reported that the plant extracts contain molecules exhibiting both strong and weak reducing power. Reducing power has a direct effect on the sizes of nanoparticles that are being developed by the biosynthesis process. Strong reducing agents result in much smaller size nanoparticles [29]. These small nanoparticles can easily penetrate within the cell membranes and biological barriers. The surface charge (ξ:zeta potential) of the AgZE nanoparticle is found to be around −14 mV (Figure 1c). The anionic charge on the silver nanoparticle helps in more colloidal stability.

### 3.4. X-ray Diffraction (XRD) Analysis

X-ray diffraction (XRD) method is used to analyze the phase purity and crystal structure of AgZE which shows broad diffraction peak from 20° to nearly 40° (2θ angle) (Figure 1d). However, no other peaks are obtained from 40–80°. A broad XRD peak corresponds to the small size of the AgZE nanoparticles [39]. The diffraction peaks are consistent with the previous patterns observed in published literature [13,33].

### 3.5. Fourier Transformed Infrared Spectroscopy (FTIR)

Both the AgZE and ZE extract are subjected to FTIR analysis to find out the presence of functional groups present in the biosynthesized AgZE. The FTIR analysis of AgZE (lower spectra) and ZE (upper spectra) are shown in Figure 1e. The major IR stretching frequencies obtained from ZE (upper spectra) extract are 3403, 2926, 1642, 1383, 1254, 1048, 919, and 609 cm^−1^, whereas the AgZE shows the IR peaks at 3421, 2924, 2853, 1740, 1634, 1545, 1512, 1463, 1383, 1248, 1163, 1042, and 537 cm^−1^. Some of the major IR peaks, observed in ZE extract are stretched in case of AgZE illustrating the involvement of active functional groups (alcohols, phenols, aldehydes and ketones, amides, etc.) in the formation of AgZE, by reduction of AgNO3 using ZE extract to generate AgZE nanoparticles. For example, the IR spectrum (upper spectra) of ZE arising at 1642 cm^−1^ may be due to the presence of aldehydes or ketones which shifts to 1634 cm^−1^ in IR spectrum (lower spectra) of AgZE, suggesting the interaction of these functional groups during the formation of AgZE [38,40]. Therefore, there is possibility in presence of polyphenols, aldehydes, ketones, and alkyl halides present within the ZE extract responsible for the formation of AgZE nanoparticles.

### 3.6. Scanning Electron Microscope (SEM)

Scanning electron microscope (SEM) helps to determine the shape and morphology of AgZE nanoparticle. The SEM image of AgZE is shown in Figure 1f,g captured from different positions, illustrating their dispersed nature with quasi-spherical form.

### 3.7. Transmission Electron Microscope (TEM)

The transmission electron microscope (TEM) analyzes the actual size, shape, and aggregation of AgZE nanoparticle. The TEM image of AgZE in Figure 1h,i referring to different magnification represents spherical, monodispersed nanoparticles with tails having sizes of less than 50 nm. The observed size of AgZE is lower in TEM than that of DLS analysis (Figure 1b). DLS analysis gives the hydrodynamic diameter of AgZE nanoparticles or nanoconjugates, whereas TEM provides the exact size of the metallic part of AgZE nanoparticles but does not include the conjugated biomolecules. As our silver nanoparticles/nanobioconjugates are biosynthesized, the surfaces of the nanoparticles are coated with various biomolecules and proteins originating from the ZE extract. Therefore, DLS provided the hydrodynamic diameter of the surface-coated biomolecules, which is always higher than that obtained from the TEM values.

### 3.8. Fluorescence Measurements

The fluorescence emission properties of AgZE are analyzed, as ZE extract solution exhibit fluorescence as per earlier report [29]. Different excitation ranges are applied from 350 nm to 650 nm, which reveals AgZE exhibit fluorescence only at green channel (380 nm excitation and 450 nm emission) as shown in Appendix A. In our earlier report, the ZE extract and AuZE exhibits fluorescence both at green (excitation 350 nm, emission 450 nm) and red (excitation 450 nm, emission 720 nm) channels [29]. However, the AgZE shows fluorescence only at green channel, which may be due to the quenching of the fluorescence molecules (present in ZE extract) during the formation of silver nanoparticles. The fluorescence property of the AgZE is utilized as bioimaging agents in mouse model.

### 3.9. Inductively Coupled Plasma Optical Emission Spectrometry (ICPOES) Analysis

To evaluate the presence of silver (Ag) in the AgZE, ICPOES analysis is performed in triplicate. The results reveal that the average concentration of silver in AgZE nanoparticles to be 0.046 mg/L when 100 µL of AgZE (from main stock) is digested in 70% HNO_3_. The final calculation revealed 1 µL of AgZE contains 0.23 µg/mL of silver. The analysis shows that the concentration of the silver remains almost same in various batches of the AgZE solution, indicating the reproducibility and yield of the reaction.

### 3.10. Cell Viability Assay

The cell viability assay using MTT reagents is used to evaluate the biocompatibility of AgZE on normal (CHO, HEK-293T, EA.hy926, and H9c2) and cancer (U-87, PANC-1, MCF-7, HeLa, and B16F10) cell lines. The AgZE does not show significant killing of normal cells at both the time points (24 h and 48 h) even at higher concentration, indicating the biocompatible nature of the nanoparticles as shown in Figure 2a–d. Furthermore, AgZE does not show significant cytotoxicity when incubated (in a dose-dependent manner) with endothelial cells (EA.hy926) at therapeutic dose (around 5 µL/mL w.r.t. cancer cells described below) and the results are presented in Appendix A along with its corresponding dose–response curve in (Appendix A). The AgZE is highly cytotoxic to the cancer cells in a dose-dependent manner as exhibited in Figure 2e–i. The percentage inhibitions of proliferation for HeLa, MCF-7, PANC-1, U-87, and B16F10 after 48 h of incubation with AgZE are 60%, 50–80%, 50–70%, ~80%, and 40–50%, respectively. The corresponding dose–response curves of the normal (CHO and HEK-293T) and cancer (HeLa, MCF-7, PANC-1, U-87, and B16F10) cells have been illustrated in Appendix A The above results show maximum cytotoxicity (around 80%) in brain cancer cells (U-87) may be due to the presence of brain targeting ability of ZE extract as published in our earlier work [29].

These results indicate that biosynthesized silver nanoparticle has a broad range of anticancer activity against various cancer cell lines in acceptance with earlier reports indicating its probable anti-proliferative nature [8,13,33]. As the AgZE is most cytotoxic to the U-87 cell lines, we have therefore calculated the IC50 in U-87 that corresponds to 0.94μg/mL (~4.1 μL) and further all in vitro assays are carried out in U-87 cell lines for detailed anticancer activity.

### 3.11. Studies on Anticancer Activity of AgZE on Glioblastoma Cells

Brain cancer is among the most malignant cancer types and results in massive mortalities worldwide. It has been reported that the unmet challenges regarding this disease include moderate prognosis as well as poor survivability rate of patients. Moreover, the plant extract ZE is brain targeting as per our earlier report, and the present data show most of the cytotoxic nature of AgZE is towards brain cancer cell lines [29]. Therefore, the detailed anticancer activity of AgZE is elucidated in the brain cancer cell line (U-87).

### 3.12. Cell Uptake Study

The cell viability assay (Figure 2 and Appendix A) of AgZE show selective cytotoxicity towards the cancer cell lines compared to normal cell lines, which is further validated by a cellular uptake study using ICPOES analysis (Appendix A). The ICPOES is carried out to check the internalization of AgZE in the normal (HEK-293T) and cancer (U-87) cell lines. The analysis illustrated that, upon treatment with AgZE, the uptake of silver is higher in cancer cell lines (U-87) in comparison to normal cell lines (HEK-293T) at 24 h, as shown in Appendix A. This causes AgZE to be more cytotoxic towards cancer cells, which can be visualized from the above mentioned MTT assay (Figure 2a–i). The study exhibits the increased uptake of AgZE in cancer cells compared to normal cells.

### 3.13. Scratch Assay

The cell migration assay is an important aspect of cancer cell progression after screening for cytotoxic assay (cell viability assay using MTT reagents) in various cancer cell lines. Earlier reports suggest that anticancer agents exhibit antimigration effects, supported by the in vitro wound scratching assay [16,32]. As all in vitro assays are carried out in U-87 (already discussed at the end of Section 3.10), the anticancer activity of AgZE is evidenced by its antimigration effect on U-87 cell line using wound scratch assay in a time dependent manner (0 h and 7 h) [16,32]. The 7 h time point decided to capture the images of wound scratch is performed as per the previously published protocol [16,32]. The results of the scratch assay demonstrate that AgZE (third row) treated cells show the least migration of cells, even after 7 h, compared to the untreated cells (first row) (Figure 3a) supporting the anticancer property of AgZE. The AgZE treatment halted the migration of cells similar to the positive control (temozolomide: TEMO, second row). The wound area is calculated using ImageJ analysis according to our previous published literature for each treatment (0 h and 7 h) and control group. The final result is the ratio of 0 h treatment group with 7 h treatment group (fold change) and compared with control group as represented graphically in Figure 3b. The results of AgZE corroborates with the earlier published report [16].

### 3.14. Transwell Migration Assay

The AgZE exhibits antimigration effect towards U-87 cells, further validated qualitatively using a transwell based migration assay as per the published literature [34,35], and data are provided in Appendix A. Generally, a transwell assay can be performed in a time-dependent manner [16]. As the wound scratching assay is performed in 7 h, the transwell assay is performed after 14 h instead of at the same time point. The representative images of migrated U-87 cells stained with crystal violet clearly observed for 14 h indicate that AgZE prevents migration of the U-87 cells as compared to the untreated control cells and cells treated with temozolomide corroborating with the results of the scratch assay (Figure 3). Temozolomide is used as positive control.

### 3.15. Cell Cycle Assay

The effect of AgZE nanoparticles on the cell cycle progression of cancer cells (U-87) is analyzed using the flow cytometry analysis. The result reflects that, for the untreated U-87 cells as shown in Figure 4a, maximum cell population is found in the G0/G1 population (corresponds to P2 gate). However, when the U-87 cells are exposed to IC50 value of AgZE, as exhibited in Figure 4c, the cell population arrest increases in sub G1 phase (corresponds to P1 gate) and G2/M (corresponds to P4 gate) phase. There is a decrease in cell population in the G0/G1 phase on AgZE treatment. All this suggests cellular damage. Similarly, temozolomide (TEMO, positive control) shows the same phases of cell cycle arrest in the sub G1 phase and G2/M phase (Figure 4b). Appendix A shows the graphical representation of the cell cycle analysis. The samples are calculated as pulled data, hence no standard deviation has been provided. As per an earlier report, treatment with the silver nanoparticles causes DNA damage and breaks in DNA structure, leading to cell cycle arrest in the sub G1 or G2/M phase which corroborates with our present AgZE results [41]. The AgZE leads to DNA damage of cells, causing an increase in the number of cells in the sub G1 (apoptotic) phase, revealing its apoptotic nature [32,33].

### 3.16. Cellular Apoptosis Assay

Apoptosis is one of the major mechanisms for cancer cell death when exposed to nanoparticles [32]. Cellular apoptosis assay is conducted by Annexin V-FITC/Propidium iodide (PI) staining through flow cytometry. The cells that do not stain for any of the two dyes are live cells populating the bottom left quadrant of the contour plot (Q3). The cells that stain with only Annexin V-FITC but not with PI are apoptotic cells (Q2: late apoptotic; Q4: early apoptotic) and the ones that stain with both the dyes are usually necrotic cells (Q1). The results indicate that, after exposure of U-87 cells to AgZE nanoparticles as illustrated in Figure 5c,c1, more cells have undergone early (at 24 h) to late apoptosis (at 30 h) in comparison to the untreated cells as in Figure 5a,a1. Moreover, temozolomide (positive control) in Figure 5b,b1 shows similar results to AgZE. The graphical representation of the analyzed data has been represented in Appendix A. The samples are calculated as pulled data, hence no standard deviation has been provided. The inference is that AgZE treated U-87 cells undergo cell cycle arrest at sub G1 and G2/M phase, leading to apoptosis as well as necrosis [8].

### 3.17. Reactive Oxygen Species (ROS) Determination

Reactive oxygen species (ROS) act dually in the survival and mortality of the cancer cells. Increase in the amount of the ROS generates oxidative stress inside the cancer cells, leading to their death by apoptosis. The U-87 cells when treated with AgZE for 5 h shows increased generation of ROS (here H_2_O_2_) by exhibiting bright green colored fluorescence as compared to untreated ones (Figure 6a) which is graphically represented in Figure 6b. The U-87 cells upon treatment with AgZE for 24 h exhibits similar results (Appendix A). As the cancer cells are metabolically more active as compared to normal cells, there is some ROS detected in untreated cells. TBHP is used as positive control. As per earlier published reports, silver nanoparticles, as well as chemotherapeutic drugs, exhibit anticancer activity through increased oxidative stress [41,42]. The ROS formation by the AgZE within the U-87 cells might be the cause of its potent anticancer activity.

### 3.18. Western Blot Analysis

The AgZE treatment alters the p53, caspase8, and STAT3 levels in the Western blot analysis of the U-87 cells, thus indicating that the anticancer activity of AgZE is probably regulating through these signaling proteins. The U-87 cells treated with AgZE illustrate upregulation of apoptotic protein caspase-8 (first blot, third column) and tumor suppressor protein, p53 (second blot, third column) with the slight upregulation of STAT3 (third blot, third column) in comparison to untreated cells (first column of each blot) as seen from the blot images of Figure 7a. The positive control temozolomide (second column of each blot) shows similar results as AgZE. The lanes of the Western blot analysis are calculated as the ratio of test proteins values to loading control protein (here GAPDH) values as percentage of normalization, quantified using ImageJ analysis have been represented graphically in Figure 7b. Even if the lanes show the same intensity, when divided, each test protein with loading control has different results. The samples are calculated as pulled data, hence no standard deviation has been provided. Both p53 and caspase 8 upregulation might lead to the death of the cancer cells as per previously published report [33].

### 3.19. Cellular Imaging by Confocal Microscopy

Fluorescent molecules are used for cell biology research as tags for proteins as well as antibodies for visualizing complex biological systems. In this regard, the in vitro fluorescence study is performed in normal (CHO) and cancer (U-87 and B16F10) cells using confocal microscopy (Figure 8 and Appendix A). The CHO (Appendix A), U-87 and B16F10 (Figure 8) cells are incubated with AgZEfor 6 h and visualized under a confocal microscope. The results indicated that the AgZE exhibit intense red fluorescence inside all the cell lines (Figure 8: Row II: treated U-87, Row IV: treated B16F10; Appendix A: Row II: treated CHO) at LDS 751 channel as compared to the untreated ones (Figure 8: Row I: untreated U-87, Row III: untreated B16F10; Appendix A: Row I: untreated CHO). Though red fluorescence is observed in all three cell lines, the cancer cell lines appear to be sick and losing morphology, indicating it is undergoing apoptosis as compared to the normal cell line. The fluorescence of AgZE may originate from single or multiple fluorescent compounds of the ZE extract having *NIR* imaging properties, which needs to be investigated further. To this context, our group also reported that Olax scandens extract also produced similar shape and size (almost) silver nanoparticles exhibiting anticancer property against melanoma cancer model [13]. However, that does not show fluorescence imaging in the *NIR* region, instead showing normal fluorescence properties. Therefore, similar shape of nanoparticles (here AgZE) does not always show similar therapeutic efficacy and *NIR* based bioimaging.

### 3.20. Chorioallantoic Membrane (CAM) Assay

Tumor angiogenesis is one of the key factors for the progression of cancer through formation of new blood vessels from the pre-existing blood vessels. The chicken egg-based CAM assay model is a well-established study for evaluating anticancer activity. Here, only the nanoparticles are incubated to visualize the effect of blood vessel inhibition by the nanoparticles for prevention of tumor angiogenesis [43,44]. Based on the in vitro anticancer activity of AgZE; the anticancer activity of AgZE is further validated by CAM assay, an in vivo approach. The chorioallantoic membrane layer in the eggs is subjected to treatment with AgZE and the effect is evaluated in a time dependent manner (0–4 h). The pictorial results shown in Figure 9a indicate a decrease in the growth of blood vessels and fold changes in the length, size, and junction of the blood vessels (analyzed by the Angioquant software) when treated with different concentrations of AgZE (2.5, 5, and 10 μL) compared to the untreated ones. The AgZE, when calculated using ICPOES, shows 1 µL = 0.23 µg/mL of silver. Therefore, 2.5 μL = 0.57 μg/mL, 5 μL = 1.14 μg/mL, and 10 μL = 2.29 μg/mL. The numerical data is graphically represented in Figure 9b. The 10 μL concentration also shows more inhibition with respect to other concentrations due to an increase in the amount of compound. These results exhibit the anticancer activity of AgZE nanoparticle synthesized from *Zinnia elegans* leaf extract. The AgZE results also corroborate with earlier published reports [16,18,45].

### 3.21. Non-Invasive Imaging of AgZE In Vivo

*NIR* based fluorescence is used as non-invasive imaging means for the diagnostic purpose. In our earlier reports, gold nanoparticles (AuZE), synthesized using ZE extract, exhibited in vivo *NIR* fluorescence at 710 nm excitation and 820 nm emission [29]. Similarly, in this study, we have synthesized AgZE from ZE extract that exhibit *NIR* bioimaging at 710 nm excitation and 820 nm emission in vivo. The fluorescence images of C57BL6/J mice after 4 h and 24 h of intraperitonial injection with ZE extract and AgZE are shown in Figure 10. The fluorescence images obtained after administration of ZE at 4 h (Figure 10: Panel I, row II, column I and II) and 24 h (Figure 10: Panel II, row II, column I and II) show accumulation in the brain and other organs. The data corroborates with the earlier published report of ZE being brain targeting in nature [29]. However, the images obtained after administration of AgZE at 4 h (Figure 10: Panel I, row III, column I and II) and 24 h (Figure 10: Panel II, row III, column I and II) exhibit increased accumulation in the brain as well as other vital organs due to the ability of nanoparticles to penetrate deeper within the tissues. The AgZE did not show *NIR* fluorescence in the solution form, which may be due to the quenching efficiency of silver. However, it shows fluorescence within the animal tissues in vivo may be due to the release of the fluorescent molecules in the blood serum from the AgZE after intraperitoneal injection in the body. The fluorescence image of the untreated animals (Figure 10: Panel I and II, row I, column I and II) of both 4 h and 24 h did not show any fluorescence except the autofluorescence in some organs.

Later, all the mice (both untreated and treated) are sacrificed, organs are removed, and fluorescence is checked through ex vivo imaging for biodistribution of ZE and AgZE at 4 h and 24 h, (i) ZE: 4 h (Figure 10: Panel I, row II, column III); 24 h (Figure 10: Panel II, row II, column III), and (ii) AgZE: 4 h (Figure 10: Panel I, row III, column III); 24 h (Figure 10: Panel II, row III, column III). The results strengthen the observations that AgZE treated mice show accumulation in organs such as the brain, spleen, kidney, colon, liver, and lung, whereas ZE treated mice show accumulation in the brain, kidney, liver, and colon both at 4 h and 24 h that is further supported by ICPOES analysis (described in the following section).

### 3.22. Biodistribution Analysis by ICPOES

The biodistribution of AgZE within the mice organs is measured by ICPOES as graphically represented in Appendix A. The results exhibit that the AgZE readily penetrate most of the organs, especially liver, heart, brain, lung, spleen, colon, and kidney, due to some phytochemicals that have efficient target ability towards these organs. The samples are calculated as pulled data, hence no standard deviation has been provided. Future studies on this observation might shed some light in identifying compounds for diagnosis and targeting of different organs.

### 3.23. Pilot Study for Biodistribution Analysis in Tumor Model

A pilot study, performed in the melanoma tumor model (using B16F10 cell line, which is mouse specific) generated in the C57BL6/J mice, showed more accumulation of the AgZE after intraperitoneal injection in the tumor site as compared to other normal organs (Appendix A). Appendix A shows the dorsal and ventral fluorescence image of mice at 4 h. Similarly, Appendix A shows the dorsal and ventral fluorescence image of mice at 24 h. Appendix A shows the representative image of the mice organs sacrificed at 24 h and bearing melanoma tumor. Appendix A illustrates enhanced fluorescence at the tumor site compared to other organs.

### 3.24. Hemolysis Assay

Hemolysis is an in vivo model where cell membrane damage causes the discharge of hemoglobin, along with other cellular components, into the plasma from the erythrocytes. Several anticancer drugs that are commercially available show toxicity towards erythrocytes, causing changes in their discoid shape and inducing hemolytic anemia [46,47]. The AgZE exhibits anticancer effect; therefore, to analyze the in vivo biocompatibility of the AgZE on the normal erythrocytes, hemolysis assay is performed using the mice RBC. The results graphically represented in Figure 11 demonstrate that AgZE (10–100 µL) is almost hemocompatible, with 10 µL (a higher dose than the w.r.t therapeutic dose of 4.1 µL) being more hemocompatible, with the potential to be used for future studies. The ZE extract depicts some amount of hemolysis at highest concentration (100 µL).

## 4. Mechanism behind Anticancer Activity of AgZE

Earlier reports explain the idea behind the anticancer activity of silver nanoparticles to release of silver ions [8,13,33]. In one such report, Lu et al. exhibited the generation of silver ions and superoxide radical following the silver nanoparticle dissolution in dissolved oxygen as per the following reaction [48]:AgNPs + O_2_ (dissolved oxygen) Ag^+^ (silver ion) + O_2_ (superoxide radical)

Previous literature on silver nanoparticles has demonstrated the release of more silver ions (both chemically and biosynthesized silver nanoparticles) in acidic environment as compared to the normal environment [13,33,36]. Following this, the silver ions are released more in the acidic microenvironment of tumor cells than in the normal cells, causing increased cytotoxicity. Therefore, the cell viability MTT assay using AgZE on various normal and cancer cell lines corroborates with the earlier reports of illustrating more cytotoxicity towards cancer cell lines [13,33]. The cell uptake study also reveals increased uptake of AgZE by cancer cell (U-87) in 24 h as compared to the normal cell line (HEK-293T).

Henceforth, the increased incorporation, as well as anticancer activity of the AgZE, is due to: (i) enhanced uptake of AgZE by the cancer cells due to their upregulated cellular metabolic rate, cell proliferation rate, and overexpression of receptors [33,49,50], (ii) enhanced permeability and retention effect (EPR) due to the leaky vasculature of tumor [51], and (iii) nanoparticles/drugs behave differently with cancer cells than normal cells due to cell type behavior [5,52]. It has been also reported that curcumin (an anticancer agent) exhibits neuronal restoration at low doses despite their anticancer activity [53]. The detailed mechanistic anticancer study of AgZE for cancer cells is under further investigation which is beyond the scope of the current study.

## 5. Discussion

The silver-based nanoparticles are used for a wide range of biomedical applications, including the cancer diagnosis and therapy, by overcoming certain limitations of the conventional treatments. Synthesis of silver nanoparticles using plant extracts (leaf, stem, root, bark, etc.) is already reported in the literature, where plant extracts work as reducing, capping, and stabilizing agents. It is well established in the published literature that plant extracts from different sources (e.g., locations, seasons, and plant parts) make nanoparticles with varying shapes, sizes, and biological activity due to presence of various active molecules present in the extract and difference in their concentration [8,13,14,15,29,31]. Moreover, the biosynthesized nanoparticles are associated with several drawbacks, such as lack of standardization, ecological imbalance of the bioresources, altered concentration of phytochemicals due to changing seasons and locations, absence of strong reducing agents in most plants, difficulty in isolation of single bioactive phytochemicals from mixed solution present in plants, and attachment of targeted phytochemicals on the surface of the nanoparticles [54]. In this present communication, the biosynthesized silver nanoparticles (AgZE) show the dual purpose of anticancer activity (without any drug) as well as *NIR* based non-invasive in vivo bioimaging. This eliminates the use of any chemotherapeutic drug or fluorescent agent for cancer theranostic applications. In our work, the ZE plant extract is used for the synthesis as well as for the stabilization of AgZE. The various molecules and fluorescent pigments, including flavonoids/polyphenols, alkaloids, terpenoids, etc., are attached to the surface of AgZE during the synthesis of the nanoparticles, making it a potential drug delivery agent. The fully developed leaves of the ZE plant are collected during the flowering stage. The reproducibility of the synthesis method is established through several batches of nanoparticle synthesis.

The optimization of reaction is carried out using a series of experiments by varying the concentration of ZE extract (**entry no. 4** in Appendix A). The optimized silver nanoparticles are thoroughly characterized by several analytical tools (XRD, DLS, SEM, TEM, FTIR, and ICPOES) and further explored for biological activities (in vitro as well as in vivo). The UV visible spectroscopy analysis of the AgZE pellet indicated the absorption peak of silver at 410 nm (Figure 1a), confirming the formation of AgZE nanoparticles (representative image: inset of Figure 1a). Appendix A indicates the time dependent variation in absorbance of different AgZE reaction sets as listed in Appendix A, with AgZE-500 (**entry no. 4**) being the optimized reaction. The hydrodynamic diameter (Figure 1b) and charge (Figure 1c) of the optimized AgZE corresponds to nearly 95 nm and around −14 mV, respectively. The hydrodynamic diameters of all AgZE reaction sets, along with the optimized product, are listed in Appendix A. The broad XRD peak of optimized AgZE (Figure 1d) indicates the crystalline nature and small size of the nanoparticles. The FTIR analysis in Figure 1e shows the different functional groups that participated in the formation of the AgZE (lower spectra). The quasi spherical surface morphology of AgZE is indicated by the SEM image in Figure 1f,g. The TEM image in Figure 1h,i exhibits AgZE having a size less than 50 nm with a spherical monodispersed shape with tails. Further, the AgZE nanoparticle itself exhibits fluorescence at 380 nm excitation and 450 nm emission in solution (Appendix A). The 0.046 mg/l amount of silver is present in 100 µL of AgZE solution is calculated using ICPOES analysis.

The biological evaluation of AgZE shows their biocompatibility to normal cell lines (CHO, HEK-293T, EA.hy926, and H9c2), but cytotoxicity to different cancer cell (U-87, PANC-1, MCF-7, and HeLa and B16F10) lines in a concentration dependent way as represented in Figure 2a–i. Furthermore, AgZE does not show significant cytotoxicity to endothelial cells (EA.hy926) at therapeutic dose (around 5 µL/mL w.r.t. cancer cells) as in Appendix A. The corresponding dose–response curve for the cell viability assays of the normal and cancer cells lines are provided in Appendix A. The IC50 of AgZE is found to be 4.1 µL (4.1 µL = 0.94 µg/mL as per ICPOES) in U-87 cell line based on the MTT assay. The cellular uptake of AgZE is more in the cancer cell (U-87) compared to the normal cell (HEK-293T) as can be seen in Appendix A due to the higher cellular metabolic rate, overexpression of receptors, and different cell type behavior of the cancer cells. As the plant extract ZE is brain targeting based on our previous report, anticancer activity of AgZE is studied in brain cancer cell line [29]. The effect of AgZE on the migration ability of U-87 cancer cell is analyzed through scratch assay. The scratch assay, as shown in Figure 3a, depicts that AgZE decreased the migration ability of U-87 cancer cells (graphically represented in Figure 3b). The transwell migration assay also revealed a similar antimigrating property of AgZE towards the U-87 cells compared to the untreated in Appendix A. The migration assay is followed by cell cycle progression analysis in Figure 4, which shows the arrest of AgZE (Figure 4c) treated U-87 cell population at sub G1 and G2/M phase, followed by a slight increase in cell population in the S phase (graphically represented in Appendix A). The cell arrest at the sub G1 phase indicates induction of apoptosis by AgZE through activation of the apoptotic proteins leading to early and late apoptotic/necrotic pathway [32,33,41]. As per earlier published reports, anticancer drugs reduced DNA synthesis where cells were arrested in S-phase transit, resulting in 60–70% of the population accumulating in S-phase in response to cytostatic circumstances. The S-phase arrest sensitizes cells to anticancer drugs or nanoparticles (here AgZE) that activate signaling for death mechanisms (such as apoptosis) followed by inhibition of survival pathways [55]. Later, the apoptosis assay results (Figure 5) show an increase in early apoptosis at 24 h followed by late apoptosis at 30 h in the AgZE treatment (Figure 5c,c1) of U-87 cells (graphically represented in Appendix A). The progression of the cells from early to late apoptosis is indicative of more DNA damage, as well as apoptosis of the cells after AgZE treatment [8,56]. The cell cycle arrest and the apoptosis are due to the ROS generation by the AgZE [57]. ROS causes an increase in oxidative stress that leads to DNA damage and cell death by apoptosis. Signal transduction pathways taking part in the induction of apoptosis are influenced by greater ROS production [41]. There are several drugs that exhibit anticancer activities through superoxide radical generation [41,42]. Therefore, AgZE treatment on U-87 for 5 h (Figure 6a) and 24 h (Appendix A) generates ROS (H_2_O_2_, one of the ROS) indicating more green colored fluorescence compared to the untreated cells. The graphical representation of the ROS assay is represented in Figure 6b.

The signaling pathways behind the anticancer activity of AgZE are determined through Western blot analysis. The cells treated with the AgZE show the upregulation of p53 (second blot, third column) and caspase-8 (first blot, third column) with the down regulation of STAT3 (third blot, third column). The results of AgZE are very similar to positive control temozolomide (second column of each blot) as shown in Figure 7a. The blots are quantified through Image J analysis as represented in Figure 7b. There is no significant change of STAT3 within all the respective groups of the U-87 cells. The caspase-8 is an important upstream mediator in the death receptor mediated apoptosis. On the other hand, p53 and its family members are the upstream regulators of caspase-8 dependent apoptosis pathway. Therefore, caspase-8 is one of the mediators of the p-53 dependent apoptosis pathway [58]. The increase in ROS by the AgZE might lead to the p53 upregulation and activation of the caspase-8 signaling pathway, leading to apoptosis [59]. For the cellular imaging, the in vitro fluorescence study using AgZE is performed in normal (CHO: as shown in Appendix A) and cancer (U-87 and B16F10: as illustrated in Figure 8) cells using confocal microscopy. The results indicate that the AgZE exhibit intense red fluorescence inside all the cell lines (Figure 8: Row II: treated U-87, Row IV: treated B16; Appendix A: Row II: treated CHO) at LDS 751 channel compared to the untreated ones.

The in vivo anticancer activity of AgZE is evaluated through its treatment on the CAM layer of the chick embryo (CAM assay: of the cancer cells by affecting tumor angiogenesis [18,45].in vivo approach), that causes a decrease in formation of the new length, size, and junctions of the blood vessels (Figure 9a). The results are quantified using Angioquant software (represented in Figure 9b). The probable anti-angiogenic property of the silver nanoparticles helps to diminish the growth.

There are reports of the silver based diagnostic approaches using several conjugated fluorophores, peptides, cell adhesion molecules, and specific antibodies for bioimaging in cancer diagnosis [60]. Our biosynthesized AgZE precludes the use of any such components, as the ZE extract contains fluorescent molecules that attach on the surface of silver nanoparticles imparting fluorescent property to them. The molecules present within the ZE extract are generally flavonoids, polyphenols, and certain phytochemicals responsible for the diagnostic properties of AgZE. Further, the fluorescent AgZE nanoform can penetrate the organs without the use of any targeting ligand. When the AgZE is injected into the C57BL6/J mice through intraperitoneal route, it shows non-invasive *NIR* fluorescence at 710 nm excitation and 820 nm emission after both 4 h (Figure 10: Panel I, row III, column I and II) and 24 h (Figure 10: Panel II, row III, column I and II) with respect to control. The animals are sacrificed to visualize the organs ex vivo, where *NIR* fluorescence helped to observe accumulation of AgZE in organs such as brain, spleen, kidney, colon, liver, and lung. The ICPOES analysis of the AgZE treated mice organs also revealed the presence of silver in the major organs with more accumulation within liver, heart, brain, lung, spleen, colon, and kidney due to occurrence of some phytochemicals that readily have efficient targeting ability towards these organs (Appendix A). Additionally, a pilot study is performed by generating melanoma tumor model in C57BL6/J mice that shows AgZE, when injected intraperitoneally, accumulates more in the tumor site as compared to other organs (Appendix A). A new molecule to be utilized within the biological system requires verification of its biocompatibility as well as in vivo toxicity. Therefore, hemocompatibility is evaluated before preceding the in vivo experiments through hemolytic assay [32]. The hemolytic assay is performed by incubating AgZE in a dose-dependent manner with mice blood, where Figure 11 demonstrates that AgZE (10–100 µL) is almost hemocompatible, with 10 µL (a higher dose than the w.r.t therapeutic dose of 4.1 µL) being a more hemocompatible dose that can be used for future studies.

It should be noted that Zinnia plant extract contains various molecules and pigments including flavonoids, polyphenols, etc., responsible for its medical and fluorescent activities [30,31]. During the synthesis of AgZE nanoparticles, some of the molecules (active or inactive molecules that can also act as reducing agents) may conjugate on the surface of AgZE. It is to be notable that the incorporation of AgZE can occur via energy dependent or independent ways. Usually, nanoparticles are incorporated inside the cells through passive diffusion [61]. However, in order to make a specific AgZE nanoparticle-based potential drug delivery system using active molecule (from extract), it should be isolated from the plant extract using Chemists from Natural Product Laboratory. The same has been initiated, however it is a long-term process, beyond the scope of the present communication.

## 6. Preclinical Approach and Future Perspective

The translation of the nanomedicine for preclinical trials needs to address several critical issues. Firstly, target specificity of the nanoparticles and drugs to the desired site of the tumor tissues is a difficult challenge. A survey reported that less than 1% of the nanoparticles being injected reach the desired site of a tumor [62]. In the current study, the AgZE acts as both a therapeutic and diagnostic agent. The cell uptake studies indicated increased uptake of AgZE by 24 h as per the ICPOES data, showing the enhanced diffusion or internalization of the nanoparticles within cancer cells without any targeting agent or chemotherapeutic drug. Secondly, the production cost, toxicity, and the therapeutic efficiency regarding nanoparticles affect the translational use of nanomedicine in the market. The development and production of AgZE is cost effective, hence it is economical in nature and can be afforded widely if clinically approved in the near future after proper biosafety evaluation in large animals. The AgZE are even non-toxic in nature, established by both in vitro (cell viability in normal cells) and in vivo (hemolytic assay). The AgZE are stable and show good dispersibility due to their even pellet dispersion. Thirdly, the other issues associated with the nanoparticles are cell penetration, immunogenicity, long term toxicity, excretion, biodegradability, administration route and dosage, pharmacokinetics, and pharmacodynamics. Taking into account all the above-mentioned issues, AgZE has the potential to be a promising cancer therapeutic, as well as diagnostic agent, in the near future after evaluating its biosafety. Long term toxicity, pharmacokinetics, pharmacodynamics, and clearance studies of the AgZE have been initiated in the laboratory, but the future of the study is beyond the scope of the current work.

## 7. Conclusions

The current study shows the development of a biocompatible, biosynthesized silver nanoparticle (AgZE) using the leaf extracts of *Zinnia elegans*. The present green chemistry approach is simple, fast, efficient, economic, ecofriendly, and non-toxic. The silver nanoparticles exhibit anticancer activity though different biological assays (in vitro). Furthermore, AgZE illustrates *NIR* based bioimaging (excitation: 710 nm, emission: 820 nm) when intraperitoneally injected in C57BL6/J mice. Interestingly, the silver nanoconjugates may be considered as cancer theranostics nanomedicine, as they work without any anticancer agent, targeting agent, or fluorescent molecule(s). The AgZE can be further utilized for preclinical applications in large animals after detailed toxicological evaluations. Considering the *NIR*-based bioimaging and medical values of ZE, as well as the anticancer activity of silver nanoparticles, the biosynthesized AgZE opens a new direction for theranostic application, with enhanced therapeutic efficacy and diagnostic applications in the near future.

## Figures and Tables

**Figure 1 cancers-13-06114-f001:**
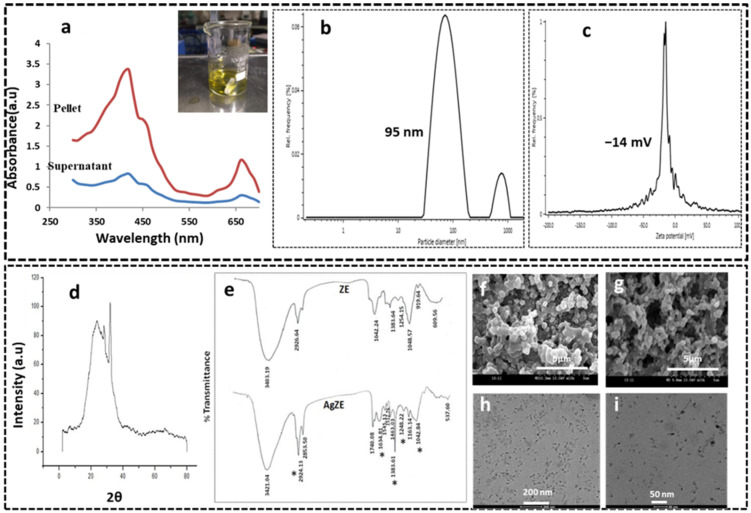
Characterizations (**a**) UV visible spectroscopy of the optimized pellet and the supernatant of AgZE; corner inset: representative image of the optimized AgZE reaction after 72 h indicating the formation of the silver nanoparticles, (**b**) hydrodynamic diameter (size) of AgZE, (**c**) charge (zeta potential) of AgZE, (**d**) XRD analysis of the AgZE showing its crystalline nature, (**e**) FTIR analysis of the ZE (top spectra) and AgZE (lower spectra) exhibiting the functional groups involved in nanoparticle synthesis, (**f**,**g**) SEM analysis of the AgZE showing its surface morphology (at magnification 5 µm for both images), and (**h**,**i**) TEM analysis of spherical shaped AgZE with minute tails. * indicates that functional groups observed in ZE extract are stretched in case of AgZE illustrating the involvement of active functional groups.

**Figure 2 cancers-13-06114-f002:**
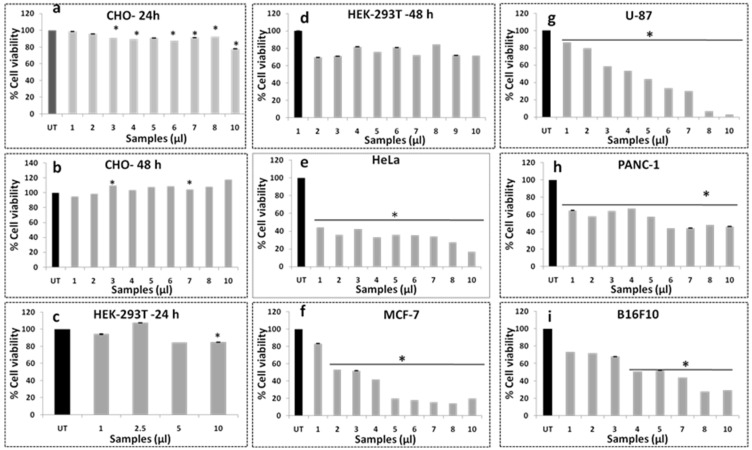
Cell viability of AgZE in normal and cancer cell line using MTT. The cell viability assay of different normal (24 h and 48 h) and cancer cell lines (48 h) incubated with AgZE using MTT reagent. The normal cell lines are: (**a**) CHO (24 h), (**b**) CHO (48 h), (**c**) HEK-293T (24 h), and (**d**) HEK-293T (48 h); the cancer cell lines are: (**e**) HeLa, (**f**) MCF-7, (**g**) U-87, (**h**) PANC-1, and (**i**) B16F10. There is dose-dependent increase in cytotoxicity of AgZE towards cancer cells compared to normal cells, indicating their anticancer activity (numerical values shows the amount of AgZE taken where 1 µL = 0.23 µg/mL of silver as per ICPOES study). Significant differences from untreated (UT) cells were observed (* *p* < 0.05).

**Figure 3 cancers-13-06114-f003:**
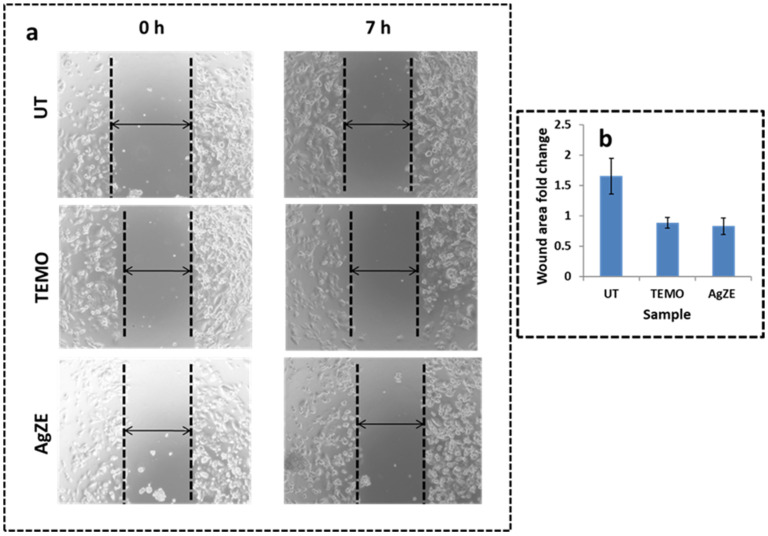
(**a**) Determination of wound healing of U-87 cell lines for untreated (first row). The AgZE inhibits the migration of cells (IC50: 4.1 μL = 0.94 μg/mL) compared to untreated cells. (**b**) Histogram representation of healing of wound area is calculated using Image J software which indicates less change in AgZE treated wound area as compared to the untreated sample. These experiments are performed thrice and represented as the mean ± SD.

**Figure 4 cancers-13-06114-f004:**
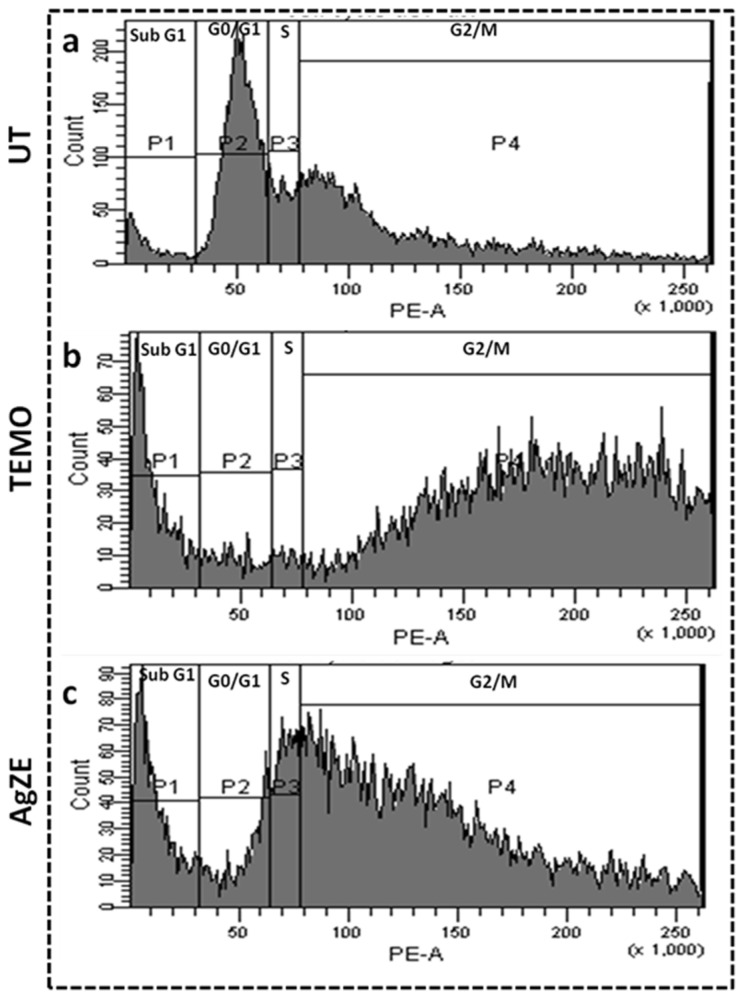
Cell cycle analysis of U-87 cells using flow cytometry, where groups are: (**a**) untreated, (**b**) temozolomide: TEMO (positive control), and (**c**) AgZE (IC50: 4.1 μL) treated at 24 h. AgZE exhibits sub G1 and G2/M phase cell cycle arrest in comparison to untreated cells showing apoptosis induced cell killing features.

**Figure 5 cancers-13-06114-f005:**
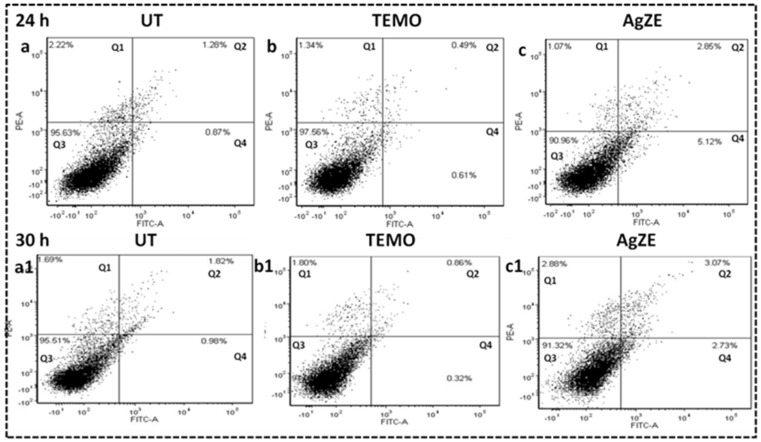
Apoptosis analysis with flow cytometry using Annexin V-FITC in U-87 cell line at 24 h (upper row) and 30 h (lower row) time points, where groups are: (**a**,**a1**) untreated: UT (**b**,**b1**) temozolomide: TEMO, positive control, and (**c**,**c1**) AgZE (IC50: 4.1 μL). AgZE treated U-87 cells exhibits increase in cell population in early apoptotic phase at 24 h which later moves to late apoptotic region at 30 h.

**Figure 6 cancers-13-06114-f006:**
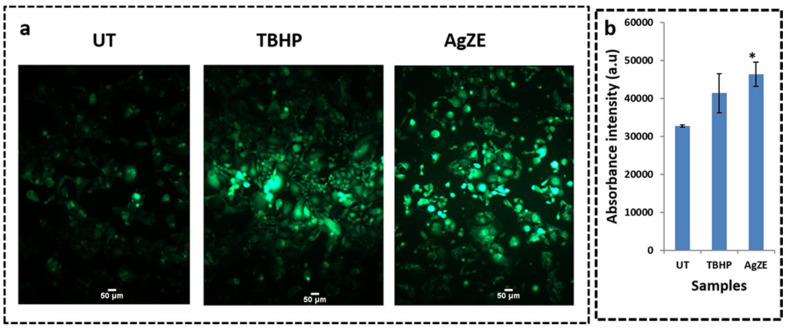
(**a**) Determination of the intracellular ROS (H_2_O_2_) production in U-87 cell line using DCFDA reagent after treatment with AgZE (IC50: 4.1 μL) and Tertiary-Butyl hydroperoxide: TBHP (positive control) for 5 h, (**b**) graphical representation of quantification of ROS generation. Higher amounts of hydrogen peroxide production is observed upon treatment with the AgZE compared to untreated (UT) cells. These experiments are performed thrice and represented as mean ± SD. Significant differences from untreated (UT) cells are observed (* *p* < 0.05).

**Figure 7 cancers-13-06114-f007:**
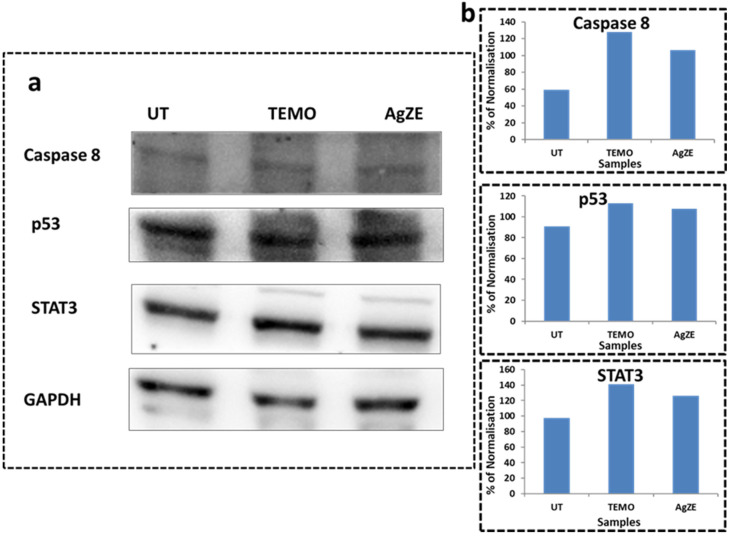
Western blot analysis of the U-87 cell lysates where the groups are untreated (first column), temozolomide: TEMO, positive control (second column), and AgZE (third column) with blots of (**a**) caspase 8 (first blot), p53 (second blot), STAT3 (third blot), and GAPDH (fourth blot), (**b**) graphical representation of quantification of Western blots using Image J analysis. Upon treatment with AgZE, U-87 cell protein lysates show upregulation of caspase-8 as well as p53 with slight upregulation of STAT3 compared to untreated samples. GAPDH is used as loading control.

**Figure 8 cancers-13-06114-f008:**
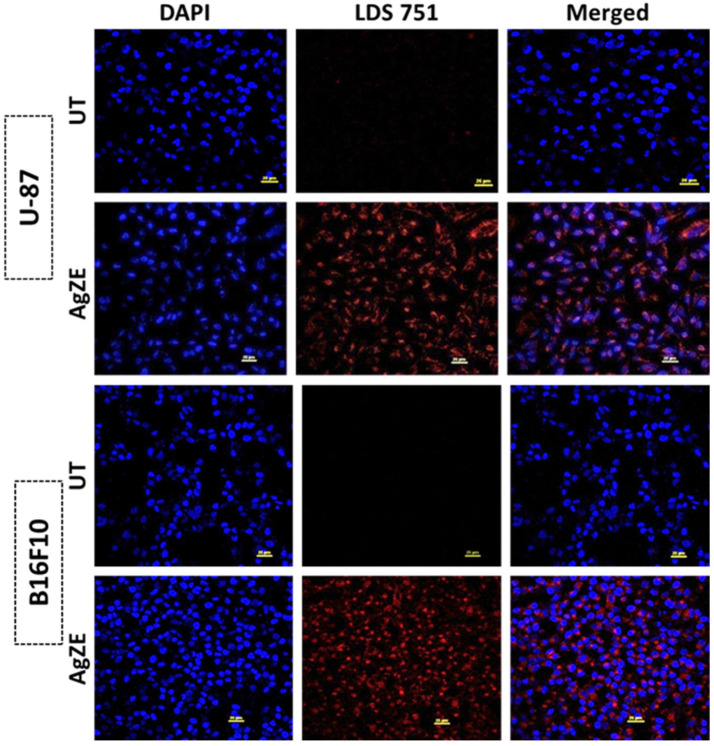
In vitro cellular uptake studies of AgZE in U-87 and B16F10 cancer cells using confocal microscopy (Row I: U-87 untreated; Row II: U-87 AgZE treated; Row III: B16F10 untreated; Row IV: B16F10 AgZE treated; Column I: Dapi; Column II: LDS 751; and Column III: Merged). The red fluorescence from AgZE-treated cells is collected under the LDS 751 channel of the confocal microscope using a 60× objective.

**Figure 9 cancers-13-06114-f009:**
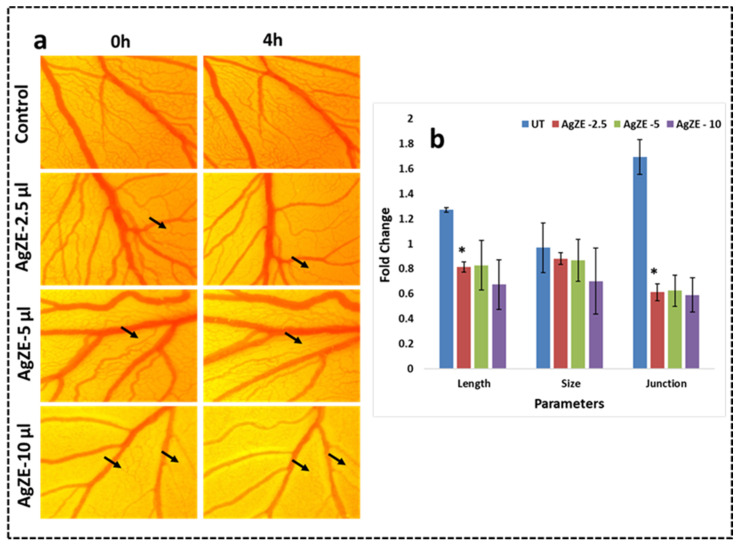
Chick embryo angiogenesis assay (**a**) Column I: 0 h; Column II: 4 h; Row I: Control (untreated); Row II: AgZE-2.5 µL; Row III: AgZE-5 µL; and Row IV: AgZE-10 µL. AgZE inhibits the growth of blood vessels that indicate the probable anticancer nature of the nanoparticles. (**b**) The images are quantified with respect to length, junction, and size using Angioquant software. The black arrows indicate the change in blood vessels of the AgZE treatment groups at 0 h and 4 h time points. These experiments are performed thrice and represented as the mean ± SD. Significant differences from UT embryo are observed (* *p* < 0.05).

**Figure 10 cancers-13-06114-f010:**
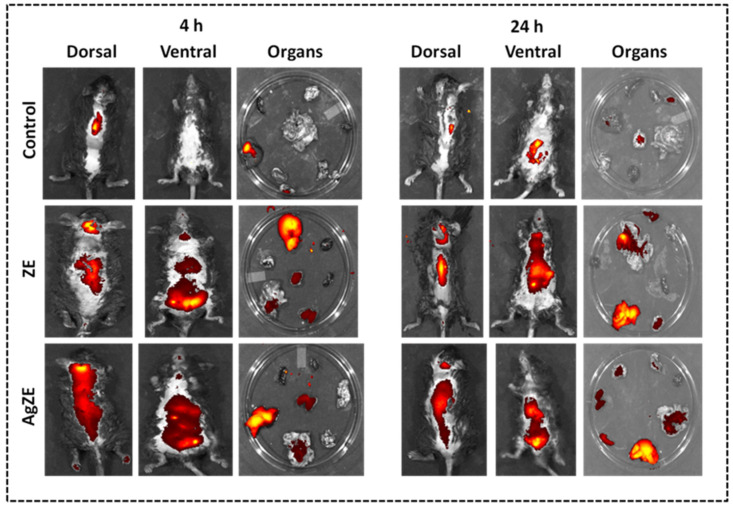
Representative images of in vivo and ex vivo biodistribution of ZE and AgZE in C57BL6/J female mice using a non-invasive in vivo imager at 4 to 24 h time points. Panel I: 4 h, Panel II: 24 h; Column I: dorsal side, Column II: ventral side, and Column III: organs of the C57BL6/J mice. Row I: Control mice; Row II: i.p. injected ZE mice; and Row III: i.p. injected AgZE mice. The images of the mice are taken at 710 nm excitation and 820 nm emission. The brain, heart, lung, liver, colon, kidney, and spleen show fluorescence upon intraperitoneal injection with ZE and AgZE, indicating their maximum distribution into those respective sites. These experiments are performed thrice.

**Figure 11 cancers-13-06114-f011:**
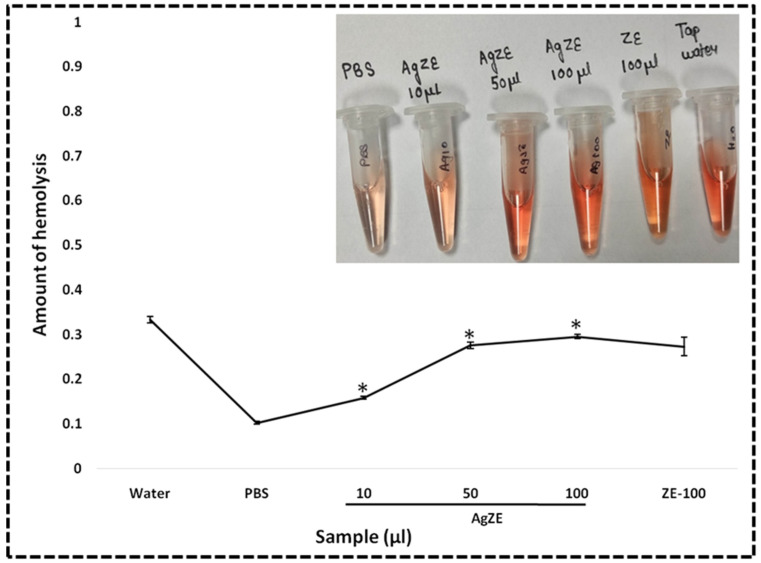
Hemolytic assay in vivo using mice blood indicating the hemocompatible nature of the AgZE (5–100 µL). The inset picture represents the amount of hemolysis (with respect to red color intensity) is more in positive control (tap water) compared to other treatments, whereas no hemolysis is observed in negative control (PBS). Numerical values indicate the AgZE pellet amount taken where 1 µL = 0.23 µg/mL of silver as per ICPOES study. These experiments are performed thrice and represented as the mean ± SD. Significant differences from negative control are observed (* *p* < 0.05).

## Data Availability

The data presented in this study are available on request from the corresponding authors. The data are not publicly available due to restrictions of the institutional IRB statement in concordance to European/German legislation on data restriction.

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
