# Peer review of "Biosynthesized Silver Nanoparticles for Cancer Therapy and In Vivo Bioimaging"

_cancers, 2021, doi:10.3390/cancers13236114_

Round 1

Reviewer 1 Report

I think we can make this a quick process since the authors have very nicely responded to the reviewer's comments, and added appropriate parts or modified others to clarify all outstanding issues. 

My main concerns were about the difficulty of standardizing nanoparticle production using plant-based extracts. But this is not related to pharmaceutical production processes, and I now agree that the level of characterization is sufficient - in fact, as good as it can be considering the circumstances (variability in plant extracts, unknown composition, etc..). 

Also a few remaining quality-related issues were solved, such as the resolution of images, and other details. The additions to the text were not many, but sufficient to address the reviewers comments. 

Therefore, I see no further options to improve the manuscript or reasons to delay the publication, and it should be accepted more or less as it is (remaining typos, etc. can be eliminated in the production process). 

Author Response

Reviewer(s)’ Comments to Author:

Reviewer # 1:

Overall comment: I think we can make this a quick process since the authors have very nicely responded to the reviewer's comments, and added appropriate parts or modified others to clarify all outstanding issues. My main concerns were about the difficulty of standardizing nanoparticle production using plant-based extracts. But this is not related to pharmaceutical production processes, and I now agree that the level of characterization is sufficient - in fact, as good as it can be considering the circumstances (variability in plant extracts, unknown composition, etc.) Also a few remaining quality-related issues were solved, such as the resolution of images, and other details. The additions to the text were not many, but sufficient to address the reviewers comments. Therefore, I see no further options to improve the manuscript or reasons to delay the publication, and it should be accepted more or less as it is (remaining typos, etc. can be eliminated in the production process).

Overall response: We are thankful to you for your nice compliments for our revised manuscript. We are also grateful to you for recommending our revised manuscript for publication in Cancers’.

Reviewer 2 Report

In this manuscript the authors describe the synthesis and the efficacy of AgZE nanoparticles in cancer treatment. After describing the simple and green process to obtain AgZE, they show its anticancer capability by using cultured cells, CAM models and mice in which they only show the AgZE distribution into the body. The authors use a lot of different techniques and obtain different data that must be presented and discussed better because, sometimes, they are not convincing.

The authors measure the cell viability of normal and cancer cells at 2 different times: 24h for normal and 48h for cancer cells. To my humble opinion, they should measure the cell viability after the same time both in normal and in cancer cells, especially if they consider that cancer cells have a shorter doubling time than normal cells.

Lanes 495-496 the authors write: “The cell migration assay is one of the important aspects of cancer cell progression after screening for cytotoxic assay (cell viability assay using MTT reagents) in U-87 cell lines”. I don’t understand why these assays are important in U-87 and not also into the other cell lines.

Why do the authors use 2 different times (7 and 14 hours) in scratch and transwell assay?

How many fields do they count in transwell assay? The authors should add the results of transwell assay (including a graphical representation with the statistical analysis derived at least from 3 experiments in triplicate).

Why do the authors perform migration and cell cycle assays only in one cancer cell line?

The authors write that in fig 4c the AgZE treatment increases the number of cells in G1 and in G2 phases compared to untreated cells and this situation indicate the existence of cellular damage. Why, if the cell proliferation is blocked, the S phase is higher in AgZE treated cells than in TEMO (positive control) and in untreated cells? Furthermore, the graph representation in fig S8 does not agree with the fig 4c (S phase).

In Western blot and relative text (lanes 675-677) the authors write: “The signaling pathways underlying the anticancer property of the AgZE (in U-87 cell lysate proteins) is demonstrated the western blot analysis which is performed to identify the key specific proteins that might be involved behind the anticancer activity of AgZE….”. Maybe, should better write: “The AgZE treatment alters the p53, caspase8 and Stat3 levels, thus indicating that…..”. Furthermore, please contrast better the caspase8, p53 and STAT3 wb (now seems that the 3 different bands are similar), indicate if, in caspase 8 WB, your antibody recognizes the full or the activated form of caspase 8 (add the molecular weight).

Lanes 730-731. The authors write: “The AgZE are incubated with CHO (Figure S11), U-87 and B16F10 (Figure 8) cells for 6 h…” usually the cells are incubated with a drug and not the opposite.

Is the AgZE incorporation energy dependent?

Lane 787: the authors indicate 2,5-5 and 10ul as AgZE concentration but these are volumes, and the concentration is unknown. Is it possible to indicate the concentration of AgZE and not only the volume?

In Par. : Pilot study for biodistribution analysis in tumor model the authors describe the effect of AgZE on mice after the inoculation of melanoma cells. First of all, the authors should describe experiments and results in a more clear and detailed way. Moreover, they should show first the tumor localization and after the AgZE localization in mice and, presumably, in cancerous area. Maybe it should be better using the data indicating the AgZE distribution in healthy mice as supplemental material and the data obtained by mice with cancer as results. If AgZE is an anticancer drug, the authors should show its effect on cancer in mice.

There are different typos and English language must be revised.

Author Response

Reviewer # 2:

Overall comment: In this manuscript the authors describe the synthesis and the efficacy of AgZE nanoparticles in cancer treatment. After describing the simple and green process to obtain AgZE, they show its anticancer capability by using cultured cells, CAM models and mice in which they only show the AgZE distribution into the body. The authors use a lot of different techniques and obtain different data that must be presented and discussed better because, sometimes, they are not convincing.

Overall response: We are grateful to you for your suggestion and advice. As per your advice and instructions, we would like to inform you respectfully that most of your critical comments, suggestions and advice were already addressed during first and second round of revisions of the manuscript. However, as per your advice and suggestions, once again we have thoroughly revised the manuscript and the changes are highlighted with yellow color.

Comment 1: The authors measure the cell viability of normal and cancer cells at 2 different times: 24 h for normal and 48h for cancer cells. To my humble opinion, they should measure the cell viability after the same time both in normal and in cancer cells, especially if they consider that cancer cells have a shorter doubling time than normal cells.

Response 1: We are thankful to reviewer’s thoughtful comments. As per the published literature, in general, the normal cells are incubated for 24 h with nanoparticles or drugs before performing cell viability assay. As per well-established literature, in this present manuscript, the normal cells (CHO & HEK) are incubated for 24 h with AgZE, whereas cancer cells (HeLa, B16F10, MCF-7, PANC-1, U87) are incubated for 48 h before cell viability assay using MTT reagents [7, 10-11, 33]. We have updated this information the experimental part of the revised manuscript.

References

  1. Pannerselvam, B.; Thiyagarajan, D.; Pazhani, A.; Thangavelu, K.P.; Kim, H.J.; Rangarajulu, S.K. Copperpod Plant Synthesized AgNPs Enhance Cytotoxic and Apoptotic Effect in Cancer Cell Lines. Processes, 2021, 9, 888
  2. Khorrami, S.; Zarrabi, A.; Khaleghi, M.; Danaei, M.; Mozafari, M. R. Selective cytotoxicity of green synthesized silver nanoparticles against the MCF-7 tumor cell line and their enhanced antioxidant and antimicrobial properties. Int. J Nanomed, 2018, 13, 8013–8024
  3. Mukherjee, S.; Kotcherlakota, R.; Haque, S.; Bhattacharya, D.; Kumar, J. M.; Chakravarty, S.; Patra, C. R. Improved delivery of doxorubicin using rationally designed PEGylated platinum nanoparticles for the treatment of melanoma. Mater. Sci. Eng. C, 2020, 108, 110375
  4. Mukherjee, S.; Kotcherlakota, R.; Haque, S.; Das, S.; Nuthi, S.; Bhattacharya, D.; Madhusudana, K.; Chakravarty, S.; Sistla, R.; Patra, C.R. Silver Prussian Blue Analogue Nanoparticles: Rationally Designed Advanced Nanomedicine for Multifunctional Biomedical Applications. ACS Biomater. Sci. Eng. 2020,6,690-704

Comment 2: Lanes 495-496 the authors write: “The cell migration assay is one of the important aspects of cancer cell progression after screening for cytotoxic assay (cell viability assay using MTT reagents) in U-87 cell lines”. I don’t understand why these assays are important in U-87 and not also into the other cell lines.

Response 2: We deeply appreciate reviewer’s critical evaluation of the manuscript. We also fully agree with reviewer’s concern. We regret for the typographical mistakes.  Accordingly, we have revised the lines (Lines 495-496) as follows:

“The cell migration assay is one of the important aspects of cancer cell progression after screening for cytotoxic assay (cell viability assay) in various cancer cell lines”

Comment 3: Why do the authors use 2 different times (7 and 14 hours) in scratch and transwell assay?

Response 3: We thank reviewer for critical comments on scratch and transwell assay. Initially, the anti-migration effect is validated by wound scratching assay after 7 h of incubation of U-87 cells (as per published literature including ours) with AgZE nanoparticles [16, 32]. It is further validated by transwell assay and the experiment is carried out as per Reviewer#1’s comments (second round of revision) [34, 35]. Generally, transwell assay can be performed in time dependent manner [16]. Since the wound scratching assay is performed in 7 h, therefore transwell assay is performed after 14 h instead of same time point.

References

  1. Limame, R.; Wouters, A.; Pauwels, B.; Fransen, E.; Peeters, M.; Lardon, F.; De Wever, O.; Pauwels, P. 2012. Comparative analysis of dynamic cell viability, migration and invasion assessments by novel real-time technology and classic endpoint assays. PLoS One. 2012, 7, e46536.
  2. Das, S.; Roy, A.; Barui, A.K.; Alabbasi, M.M.A.; Kuncha, M.; Sistla, R.; Sreedhar, B.; Patra, C.R. Anti-angiogenic vanadium pentoxide nanoparticles for the treatment of melanoma and their in vivo toxicity study. Nanoscale. 2020,12,7604-7621.
  3. Deng, G.; Zhou, F.; Wu, Z.; Zhang, F.; Niu, K.; Kang, Y.; Liu, X.; Wang, Q.; Wang, Y.; Wang, Q. Inhibition of cancer cell migration with CuS@ mSiO(2)-PEG nanoparticles by repressing MMP-2/MMP-9 expression. Int. J Nanomed. 2017, 13, 103-116.
  4. Kovács, D.;Igaz, N.; Marton, A.; Rónavári, A.; Bélteky, P.; Bodai, L.; Spengler, G.; Tiszlavicz, L.; Rázga, Z.; Hegyi, P.; Vizler, C.; Boros, I. M.; Kónya, Z.; Kiricsi, M. Core-shell nanoparticles suppress metastasis and modify the tumour-supportive activity of cancer-associated fibroblasts. J Nanobiotechnol. 2020, 18, 18.

Comment 4: How many fields do they count in transwell assay? The authors should add the results of transwell assay (including a graphical representation with the statistical analysis derived at least from 3experiments in triplicate).

Response 4: For the transwell assay three to four fields have been randomly captured from each 24 well. Since the cells appear to have been clumped in the untreated wells, we are unable to measure them quantitatively. Therefore, we are providing qualitative data in Supporting Information (Figure S7).

Comment 5: Why do the authors perform migration and cell cycle assays only in one cancer cell line?

Response 5: We are thankful to reviewer’s thoughtful comments. Initially cytotoxic assays are performed (screened) in various cancer cells (HeLa, B16F10, MCF-7, PANC-1, U87) (Figure 2) after incubation with AgZE nanoparticles and the results shows that AgZE being most cytotoxic to U-87 cells. Moreover, our earlier report suggests the plant extract ZE is brain targeting as per our earlier report and the present data shows most cytotoxic nature of AgZE towards brain cancer cell lines (U-87) [29]. Therefore, the detailed anticancer activity (all in vitro assays) of AgZE is elucidated in the brain cancer cell line (U-87). The explanation is already provided at the end of Section 3.10.

Comment 6: The authors write that in fig 4c the AgZE treatment increases thenumber of cells in G1 and in G2 phases compared to untreatedcells and this situation indicate the existence of cellular damage.Why, if the cell proliferation is blocked, the S phase is higher in AgZE treated cells than in TEMO (positive control) and inuntreated cells? Furthermore, the graph representation in fig S8 does not agree with the fig 4c (S phase).

Response 6: We are thankful to reviewer’s thoughtful suggestion. As per earlier published reports, anticancer drugs reduced DNA synthesis where cells are arrested in S-phase transit, resulting in 60-70% of the population accumulated in S-phase in response to cytostatic circumstances. The S-phase arrest sensitizes cells to anticancer drugs or nanoparticles (here AgZE) that activate signaling for death mechanisms (such as apoptosis) followed by inhibition of survival pathways [54].

             The scale of Y-axis in Figure S8 corresponds for UT is from 0-200 and for AgZE is for 0-90. Therefore, it seems that the cell population of S phase is more in AgZE than UT but in actual through data analysis by FCS software the cell population in S phase of UT is more than AgZE.                          

References

  1. Lee, C.C.; Lin, M.L.; Meng, M.; Chen, S.S. Galangin induces p53-independent S-phase arrest and apoptosis in human nasopharyngeal carcinoma cells through inhibiting PI3K–AKT signaling pathway. Anticancer Res, 2018, 38, 1377-1389.

Comment 7: In Western blot and relative text (lanes 675-677) the authors write: (i) “The signaling pathways underlying the anticancer property of the AgZE (in U-87 cell lysate proteins) is demonstrated the western blot analysis which is performed to identify the key specific proteins that might be involved behind the anticancer activity of AgZE….”

. Maybe, should better write: “The AgZE treatment alters the p53, caspase8 and Stat3 levels, thus indicating that…..”. (ii) Furthermore, please contrast better the Caspase 8, p53 and STAT3 wb (now seems that the 3 different bands are similar), (iii) indicate if, in caspase 8 WB, your antibody recognizes the full or the activated form of caspase 8 (add the molecular weight).

Response 7: First of all, we would like to sincerely thank reviewer for giving valuable time to critically evaluate the manuscript.

  1. i) As per the reviewer’s comment we have updated the line in the revised manuscript.
  2. ii) The contrasts of the western blot bands (already done) increased and revised as per Reviewer#1’s comments (during second round of revision).

iii) The anti-caspase 8 recognizes full caspase 8 of the western blot band.

Comment 8: Lanes 730-731. The authors write: “The AgZE are incubated with CHO (Figure S11), U-87 and B16F10 (Figure 8) cells for 6 h…”usually the cells are incubated with a drug and not the opposite.

Response 8: We are extremely sorry for such typographical mistake. As per the reviewer’s comment we have updated the line in the revised manuscript. We have also revised the same all over the manuscript.

Comment 9: Is the AgZE incorporation energy dependent?

Response 9: We are thankful to reviewer’s thoughtful comments. The incorporation of AgZE can occur through energy dependent or independent way. Usually, nanoparticles are incorporated inside the cells through passive diffusion [61]. As per the published literature, we assume that the enhanced uptake of AgZE by the cancer cells (i) may be due to their upregulated cellular metabolic rate, cell proliferation rate and overexpression of receptors [33, 49, 50], (ii) enhanced permeability and retention effect (EPR) because of the leaky vasculature of tumor,[51] (iii) nanoparticles/drugs behave differently with cancer cells than normal cells due to cell type behavior [5, 52]. Moreover, the ZE plant extract is used for the synthesis of AgZE has various molecules and fluorescent pigments including flavonoids/polyphenols, alkaloids, terpenoids, etc. are attached to the surface of AgZE during the synthesis of the nanoparticles, making it a potential drug delivery agent.

References

  1. Senapati, S.; Mahanta, A.K.; Kumar, S.; Maiti, P. Controlled drug delivery vehicles for cancer treatment and their performance. Signal Transduct. Target. Ther. 2018,3,7.
  2. Mukherjee, S.; Kotcherlakota, R.; Haque, S.; Das, S.; Nuthi, S.; Bhattacharya, D.; Madhusudana, K.; Chakravarty, S.; Sistla, R.; Patra, C.R. Silver Prussian Blue Analogue Nanoparticles: Rationally Designed Advanced Nanomedicine for Multifunctional Biomedical Applications. ACS Biomater. Sci. Eng. 2020,6,690-704.
  3. Patra, C.R.; Bhattacharya, R.; Wang, E.; Katarya, A.; Lau, J.S.; Dutta, S.; Muders, M.; Wang, S.; Buhrow, S.A.; Safgren, S.L.; Yaszemski, M.J. Targeted delivery of gemcitabine to pancreatic adenocarcinoma using cetuximab as a targeting agent. Cancer Res. 2008,68,1970-1978
  4. Wang, Y.H.; Israelsen, W.J.; Lee, D.; Vionnie, W.; Jeanson, N.T.; Clish, C.B.; Cantley, L.C.; Vander Heiden, M.G.; Scadden, D.T. Cell-state-specific metabolic dependency in hematopoiesis and leukemogenesis. Cell. 2014,158,1309-1323.
  5. Kalyane, D.; Raval, N.; Maheshwari, R.; Tambe, V.; Kalia, K.; Tekade, R.K. Employment of enhanced permeability and retention effect (EPR): Nanoparticle-based precision tools for targeting of therapeutic and diagnostic agent in cancer. Mater Sci Eng C Mater Biol Appl. 2019,98,1252-1276.
  6. Tiwari, S.K.; Agarwal, S.; Seth, B.; Yadav, A.; Nair, S.; Bhatnagar, P.; Karmakar, M.; Kumari, M.; Chauhan, L.K.S.; Patel, D.K.; Srivastava, V. Curcumin-loaded nanoparticles potently induce adult neurogenesis and reverse cognitive deficits in Alzheimer’s disease model via canonical Wnt/β-catenin pathway. ACS Nano. 2014,8,76-103.
  7. Foroozandeh, P.; Aziz, A.A. Insight into Cellular Uptake and Intracellular Trafficking of Nanoparticles. Nanoscale Res. Lett.2018, 13,339

Comment 10: Lane 787: the authors indicate 2,5-5 and 10ul as AgZE concentration but these are volumes, and the concentration is unknown. Is it possible to indicate the concentration of AgZE and not only the volume?

Response 10: The AgZE when calculated using ICPOES shows 1µl= 0.23 µg/ml of silver. Therefore, 2.5 μl =0.57 μg/ml, 5 μl =1.14μg/ml, 10 μl =2.29μg/ml.

Comment 11: In Par.: Pilot study for biodistribution analysis in tumor model

The authors describe the effect of AgZE on mice after the inoculation of melanoma cells. First of all, the authors should describe experiments and results in a more clear and detailed way. Moreover, they should show first the tumor localization and after the AgZE localization in mice and, presumably, in cancerous area. Maybe it should be better using the data indicating the AgZE distribution in healthy mice as supplemental material and the data obtained by mice with cancer as results. If AgZE is an anticancer drug, the authors should show its effect on cancer in mice.

Response 11: We deeply appreciate reviewer’s constructive comment.

The AgZE distribution in healthy mice is already shown in Figure 10 and it depicts only the diagnostic properties of AgZE in the NIR region. Additionally, uptake of the Ag in different organs (liver, heart, brain, lung, spleen, colon and kidney) detected by ICPOES and data presented in Figure S12.

            In this present communication, we demonstrate that the biosynthesized silver nanoparticles (AgZE) show the dual purpose of anticancer activity (without any drug) as well as NIR based non-invasive in vivo bioimaging.

      Apart from that, we have initiated the detailed anti-cancer activity of the nanoparticles (AgZE) in preclinical mouse model along with long term toxicity, pharmacokinetics, pharmacodynamics and clearance in our laboratory. But these studies are in progress and beyond the scope of the current work.

Comment 12: There are different typos and English language must be revised.

Response 12: As per the reviewer’s comment we have updated the revised manuscript and edited by professional English.

Finally, we are grateful to the Editor and reviewers for their critical comments, suggestions and advice that enormously helped us to improve the quality of the revised manuscript.

We hope the revised manuscript (at present form) is now suitable for publication in ‘Cancers’

Round 2

Reviewer 2 Report

Dears Authors, please allow me to tell you an important message. Scientific research is not done to publish but to discover. When you submit a manuscript, you have to accept the comments because your work can ameliorate and can be useful to other researchers. When a reviewer ask you to show something, you have to demonstrate it with facts and not with suppositions (in literature..... Only in this way you can be sure of you data.

For example, if I tell you: "The authors measure the cell viability of normal and cancer cells at 2 different times: 24 h for normal and 48h for cancer cells. To my humble opinion, they should measure the cell viability after the same time both in normal and in cancer cells, especially if they consider that cancer cells have a shorter doubling time than normal cells." is not because I'm crazy but because a slight effect after 24h treatment can  increase after other24 hours. In this way, the cell viability and the cytotoxic effects can increase and also influence proliferation and survival of normal cells. 

The same is for different other comments

Author Response

This manuscript is a resubmission of an earlier submission. The following is a list of the peer review reports and author responses from that submission.

Round 1

Reviewer 1 Report

In this manuscript the authors want to demonstrate the efficiency of AgZE in cancer therapy. They illustrate how synthetize this compound and would demonstrate the efficiency of AgZE for contrasting proliferation, migration and angiogenesis in different cancer cells. The work could be considerably ameliorated.

In cell migration assay the AgZE treatment seems to elicit no effects. Maybe the migration time is not enough, or the authors should use a more sensitive test such as transwell assay. How it’s possible that after 7h AgZE treatment the wound width is larger than in untreated cells? How did the authors calculate the wound area?

The authors assert that AgZE treatment induces cell apoptosis, but they don’t demonstrate this (for example by using different markers).

Lanes 514-526 the authors illustrate the signaling pathway used by AgZE to inhibits cancer cell growth and migration. I think that they don’t demonstrate a signaling pathway but only show the total quantity of different proteins. Furthermore in all the wb the lanes seems to have the same intensity. Again, the authors do not indicate the stimulation time in this experiment.

In chick embrio angiogenesis assay we (maybe) can observe a slight inhibition of vessel formation in presence of 10ul treatment.

Author Response

Reviewer(s)’ Comments to Author:

Reviewer 1:

Overall comment: In this manuscript the authors want to demonstrate the efficiency of AgZE in cancer therapy. They illustrate how synthetize this compound and would demonstrate the efficiency of AgZE for contrasting proliferation, migration and angiogenesis in different cancer cells. The work could be considerably ameliorated.

Overall response: We are grateful to you for giving us the opportunity to revise the manuscript for publication in the esteemed journal Cancers’. Based on your advice and instructions, we have revised the manuscript carefully. We hope that you will now recommend the revised manuscript suitable for publication in ‘Cancers’.

Comment 1:

  1. i) In cell migration assay the AgZE treatment seems to elicit no effects. Maybe the migration time is not enough, or the authors should use a more sensitive test such as transwell assay.
  2. ii) How it’s possible that after 7h AgZE treatment the wound width is larger than in untreated cells?

iii) How did the authors calculate the wound area?

Response 1: We are thankful to the reviewer’s thoughtful comments.

  1. i) The cell migration assay is one of the important aspects of cancer cell progression after screening for cytotoxic assay (cell viability assay using MTT reagents) in U87 cell lines. Earlier reports suggest that anticancer agents exhibit antimigration effects, supported by in vitro wound scratching assay [16, 32]. The anticancer activity of AgZE is evidenced by its antimigration effect on U-87 cell line using wound scratch assay. The 7 h time point decided to capture the images of wound scratch is performed as per the previously published protocol [16, 32].

To perform the transwell assay (recommended by Reviewer #1), we have enquired with the distributors for the reagents. Unfortunately, they are unable to deliver them in very short time (within deadline October 7-2021) due to COVID pandemic situation.

  1. ii) The AgZE (anticancer agent) prevents the U-87 cells from migrating towards each other as compared to control. Also, at 7 h there are smaller numbers of viable U-87 cells due to the AgZE treatment (since it is cytotoxic) compared to untreated cells leading to larger wound width.

iii) The wound area is calculated using ImageJ analysis according to our previous published literature for each treatment (0 h & 7 h) and control group. The final result is the ratio of 0 h treatment group with 7 h treatment group (fold change) and compared with control group.

References:

  1. Buttacavoli, M.; Albanese, N.N.; Di, Cara. G.; Alduina, R.; Faleri, C.; Gallo, M.; Pizzolanti, G.; Gallo, G.; Feo, S.; Baldi, F.; Cancemi, P. Anticancer activity of biogenerated silver nanoparticles: an integrated proteomic investigation. Oncotarget. 2017,9,9685-9705.
  2. Das, S.; Roy, A.; Barui, A.K.; Alabbasi, M.M.A.; Kuncha, M.; Sistla, R.; Sreedhar, B.; Patra, C.R. Anti-angiogenic vanadium pentoxide nanoparticles for the treatment of melanoma and their in vivo toxicity study. Nanoscale. 2020,12,7604-7621.

Comment 2: The authors assert that AgZE treatment induces cell apoptosis, but they don’t demonstrate this (for example by using different markers).

Response 2: We thank the reviewer for valuable suggestion. The standard techniques for apoptosis reported in previously published literature are cell cycle as well as Annexin-VFITC apoptosis (using FACS analysis) and western blot analysis (regulation of apoptotic proteins: e.g caspase 8) [32,55]. Therefore, we have performed these analyses to support our hypothesis of AgZE being anticancer through their apoptotic nature. The detailed explanation has been provided in the revised main manuscript.

Apart from the above well-known apoptosis assays, TUNEL assay and other markers using western blot analysis (caspase proteins, Bax, Bak, Bid, etc.) could be incorporated in the manuscripts as supporting data.

References:

  1. Das, S.; Roy, A.; Barui, A.K.; Alabbasi, M.M.A.; Kuncha, M.; Sistla, R.; Sreedhar, B.; Patra, C.R. Anti-angiogenic vanadium pentoxide nanoparticles for the treatment of melanoma and their in vivo toxicity study. Nanoscale. 2020,12,7604-7621.

55.Liu, J., Uematsu, H.; Tsuchida, N.; Ikeda, M.A. Essential role of caspase-8 in

p53/p73-dependent apoptosis induced by etoposide in head and neck carcinoma cells. Mol. Cancer, 2011, 10, 95.

Comment 3:

  1. i) Lanes 514-526 the authors illustrate the signaling pathway used by AgZE to inhibits cancer cell growth and migration. I think that they don’t demonstrate a signaling pathway but only show the total quantity of different proteins.
  2. ii) Furthermore in all the wb the lanes seems to have the same intensity.

iii) Again, the authors do not indicate the stimulation time in this experiment.

Response 3: We are thankful to the reviewer’s critical evaluation.

  1. i) As per the comment, the western blot analysis is performed to identify the key specific proteins that might be involved behind the anticancer activity of AgZE towards the U-87 cell line. The upregulation and downregulation of proteins indicates the probable signaling pathway through which AgZE is functioning. We have demonstrated that the probable anticancer activity of AgZE is due to STAT3 inhibited, p53 dependent caspase-8 mediated apoptosis pathway.

  1. ii) The lanes of the western blot analysis are calculated as the ratio of test proteins values to loading control protein (here GAPDH) values as percentage of normalization, quantified using ImageJ analysis. Even if the lanes show same intensity when each test protein is divided with loading control the results are different.

% of Normalization: values of test protein/values of loading control protein

iii)The stimulation/incubation time (18 h) for the AgZE treatment in U-87 cells have been already mentioned supporting information file and the same has been updated in main manuscript.

Comment 4: In chick embrio angiogenesis assay we (maybe) can observe a slight inhibition of vessel formation in presence of 10ul treatment

Response 4: We appreciate reviewer’s thoughtful evaluation. Tumor angiogenesis is one of the key factors for the progression of cancer through formation of new blood vessels from the pre-existing blood vessels. The chicken egg-based CAM assay model is a well established study for evaluating anticancer activity. Here, only the nanoparticles are incubated to visualize the effect of blood vessel inhibition by the nanoparticles for prevention of tumor angiogenesis [41-42]. The CAM assay results as shown in Figure 8a indicates a decrease in the growth of blood vessels (w.r.t. length, size and junction) when treated with different concentrations of AgZE (2.5, 5 and 10 μl) compared to the untreated ones and analyzed by the Angioquant software. The 10 μl concentration also shows more inhibition with respect to other concentration due to increase in the amount of compound.

References

41.Vu, B.T.; Shahin, S.A.; Croissant, J.; Fatieiev, Y.; Matsumoto, K.; Doan, T.L.H.; Yik, T.; Simargi, S.; Conteras, A.; Ratliff, L.; Jimenez, C.M. Chick chorioallantoic membrane assay as an in vivo model to study the effect of nanoparticle-based anticancer drugs in ovarian cancer. Sci. Rep.2018, 8, 8524.

42.Merlos Rodrigo, M.A.; Casar, B.; Michalkova, H.; Jimenez Jimenez, A.M.; Heger, Z. ; Adam, V. Extending the Applicability of In Ovo and Ex Ovo Chicken Chorioallantoic Membrane Assays to Study Cytostatic Activity in Neuroblastoma Cells. Front. Oncol. 2021, 11, 707366.

Finally, we are grateful to the Editor and reviewers for their critical comments, suggestions and advice that enormously helped us to improve the quality of the revised manuscript.

Hope the manuscript is now suitable for publication in ‘Cancers’

Reviewer 2 Report

I have aproblem with the lack of standardization of almost any of the approaches described here. Where to get started? 

1) The preparation of silver nanoparticles is not in any way standardized, and close to nothing is known about the chemical reactions? How can the authors be sure there are SPECIFIC reactions when using extracts from Zinnia elegans? Are different nanoparticles fromed when extracts from other plats are used? And would these nanoparticles (as described in reference 12, here generated with extracts from Olax scandens) have different anti-cancer activities? Why arent any other "biosynthesized" nanoparticles used and compared in this study? I would have expected such controls and comparisons as a central element of a manuscript that aims at a more thorough characterization of the anti-cancer effects, and their mechanisms. 

The characterization of the silver NPs is not described, instead there is a reference to the characterization of gold NPs instead. I dont think this suffices for the purpose of this manuscript: the reader would really want to see the structure, shape, sizes etc. of the silcer NPs that are generated secifically (?) when the Zinnia extracts are used. Why Zinnia? How has this been selected, was there a screening process? That, however, is provided nowhere. I agree with the authors that maybe the structure of the NPs formed with Zinnia extracts is similar and comparable to the ones generated with other plants. But - its simply not demonstrated how and if these NPs are different, have more beneficial properties that others, etc. Isnt it possible that ANY plant extracts will contain sufficient reducing biomolecules that will resiult in the reduction of silver ions and precipitation of nanoparticles that have a mixed and probably very heterogeneous content? It is hinted in the manuscript to the differential reductive potential of plant extracts, leading to different sized NPs; but this may be a rather generic observation and all plant extracts are likely to contain reductive reagents (polyphenoles etc.). 

Therefore, better characterization of this material in comparison to the previously published NPs, and other methods of their generation is an absolute MUST before this manuscript can be published. It almost makes no sense to even go through the results before this isnt properly done. There is no way other researchers could follow up on this without additional information. 

Otherwise, the NPs used here may of course have potential. Some of the experiments performed are conclusive, others less. For example, whats the point of performing hemolysis assays? How is this connected to the anti-cancer activities observed? 

The optical properties of these nanoparticles are interesting, especially if it should be true that there are differences in cytotoxicity between "normal" (HEK-293, CHO) and various cancer cells. But, is there also a potential explanation for this? Are similar properties observed with other NPs generated in a similar way? r is this really something outstanding that is specific for the Zinnia-NPs? These kind of controls are missing, and would make the story more interesting and relevant. By the way, does/response curves are also missing, or an estaimate to EC50 values. The concentrations used should cover a borader range - this is illustrated with HeLa cells, where there is no difference between 1 and 10 ul of reagent.  There should be more exponential (base 2) increasees in the acive concentrations used. And the concentrations should be measured more precisely: The quantification "ul" is not precise and insufficient for this type of activity assays. 

The cellular uptake assays are also poorly described, and the methods are soewhat cryptic: "The samples are calculated as pulled data hence no standard deviation has been provided" (line 395). This needs to be improved. 

Some other experimental approaches are convincing and informative, such as the scratch wound assays, the cell cycle/apoptosis analyses, and the ROS assays - pointing to rather generalized cytotoxicity, maybe. This specifically applies to the experiments deon with thre U-87 cell line; here the used controls and experimental conditions appear appropriate and sufficcient. Nevertheless, it may be an important issue (if the role of these NPs is further explored) to add more non-trnaformed, normal cells into these various assays formats, and validate there is a true difference in sensitivity between normal and cancer cells. 

Concerning the Western blots, and the quantification: I cannot see and significant differences? Why have these genes be selected to be investigated? (I can kind of understand caspase 8). 

How does the growth of bllod vessels in CAM assay relate to the previous experiments done with cancer cell lines? There doesnt appear to be a logical  connection? (Leaky and defective bllod vessels in cancers?). Plus, is there any significant difference? I am not convinced there is. 

The in vivo imaging then appears rather relevant and conclusive again; although the quality of the images can be improved. It wozkd all make much more sense, however, it mice with tumor xenografts woud have been used (for example, with the U-87 cell line) and there would have been significant incorpoation or enrichment of the NPs in comparison to any other tissues. That, however, has not been done. 

The paragraph on possible reasons for the cancer-specific action of the NPs is all speculation, none of this has been proved and demonstrated by any of the experiments shown in the manuscript. This section is not helping. 

minor issues: 

English language use is sometimes a bit "funny", and requires moderate changes. There some odd typos in some sentences, like missing words and articles or prepositions etc., or "its" instead of "their". There are many of these, none really terrible but it does affect the overall impression. 

The discussion is largely a recapitulation of the results, and not really a discussion highlighting pros and cons of the study, or possible linke to existing literature by others. That can also be improved. 

Author Response

Reviewer(s)’ Comments to Author:

Reviewer 2:

Overall Response: We are grateful to Reviewer #2 for giving us the opportunity to thoroughly revise the manuscript before publication in Cancers’. Based on your advice and instructions, we have revised the manuscript very carefully. We have also carried outsome additional new experiments (cell viability assay in B16F10 cells, in vitro confocal imaging in normal CHO cells and cancer cells: U-87 and B16F10, in vivo NIR based imaging in melanoma tumor model in C57BL6/J mice) and repeated some earlier experiments (FTIR, SEM and ICPOES) that clearly demonstrated our hypothesis (cancer therapy and in vivo imaging of AgZE). We hope that Reviewer #2 will now recommend the revised manuscript for publication.

Comment 1:

  1. i) The preparation of silver nanoparticles is not in any way standardized, and close to nothing is known about the chemical reactions?
  2. ii) How can the authors be sure there are SPECIFIC reactions when using extracts from Zinnia elegans?

iii) Are different nanoparticles fromed when extracts from other plats are used? And would these nanoparticles (as described in reference 12, here generated with extracts from Olax scandens) have different anti-cancer activities?

  1. iv) Why arent any other "biosynthesized" nanoparticles used and compared in this study? I would have expected such controls and comparisons as a central element of a manuscript that aims at a more thorough characterization of the anti-cancer effects, and their mechanisms.

Response 1: We are thankful to the reviewer’s critical comments.

  1. i) Synthesis of silver nanoparticles using plant extracts (leaf, stem, root, bark, etc) is already reported by numerous literature where plant extract works as reducing, capping as well as stabilizing agent. Moreover, the biosynthesized nanoparticles are associated with several drawbacks such as lack of standardization, ecological imbalance of the bioresources, altered concentration of phytochemicals due to changing seasons and locations, absence of strong reducing agents in most plants, difficulty in isolation of single bioactive phytochemicals from mixed solution present in plants, attachment of targeted phytochemicals on the surface of the nanoparticles, etc [52].

In this present study, silver nanoparticles (AgZE) are synthesized according to published protocols using ethanolic (ZE) leaf extract where it acts as a reducing, stabilizing and capping agent. The ZE extract contains different phytochemicals, polyphenolic/alcoholic compounds and aldehydes/ketones that help in the reduction of Ag2+ (in AgNO3) to Ag0 as per the reported literature [8,29,31]. The optimization of reaction is carried out using a series of experiments by varying the concentration of ZE extract (Entry No.4 in Table S1). The optimized silver nanoparticles are thoroughly characterized by several analytical tools (XRD, DLS, SEM, TEM, FTIR and ICPOES). We have also included updated SEM, FTIR and ICPOES data in the revised manuscript.

  1. ii) We fully agree with reviewer. It is reported that Zinnia plant extract contains various molecules and pigments including flavonoids, polyphenols, etc. responsible for its medical and fluorescent activities [30, 31]. During the synthesis of AgZE nanoparticles, some of the molecules (active or inactive molecule that can also act as reducing agent) may conjugate on the surface of AgZE. However, in order to make AgZE nanoparticles based specific potential drug delivery system using active molecule (from extract), it should be isolated from the plant extract using Chemists from Natural Product Laboratory. We have initiated the same, however it is a long term process and beyond the scope of the present communication.

                  However, if the source or region of Zinnia elegans extract is changed we have to standardize the reaction since with seasonal and regional changes the plant protein content as well as reducing properties change.

iii) Yes, different nanoparticles are formed from different plant extracts. It is well established by published literature that plant extract from different sources (e.g. locations, seasons and plant parts) make nanoparticles with varying shape, size and biological activity due to presence of various active molecules present in the extract and difference in their concentration [8, 13-15, 29,31].

Our group has already developed several biosynthesized nanoparticles (silver, gold -based) using different source of plant leaf extract and demonstrated their biological properties in various ways. For example, biosynthesized silver nanoparticles using Olax scandens shows the potential application as drug delivery system along with fluorescence properties anticancer as well as antibacterial activity [13]. Gold nanoparticles synthesized using Eclipta alba shows biocompatibility and would be useful for drug delivery applications [14]. On the other hand, gold nanoparticles synthesized using Hamelia patens extract show pro-angiogenic properties [15]. Biosynthesized gold nanoparticles (AuZE) obtained by the reduction of gold salts using extracts of Zinnia elegans exhibit fluorescence at both green (emission: 450 nm; excitation: 350 nm) and red region (emission: 720 nm; excitation: 450 nm) [29]. However, AuZE do not exhibit anticancer activity unless we use any chemotherapeutic drug.

Therefore, different nanoparticles produced with different plant extracts shows varying biological properties due to presence of various active phytochemicals and reducing agents.

  1. iv) The current study focuses on the biosynthesized silver nanoparticles (AgZE) that demonstrate the anticancer activity and NIR -based fluorescence bioimaging. We have also used temozolomide (brain cancer drug) as positive control and compared the results with AgZE.

   Our group already synthesized biosynthesized gold nanoparticle (AuZE different from AgZE) using the same plant extract and demonstrates the NIR-based fluorescence bioimaging [29]. These nanoparticles (AuZE) do not exhibit anticancer activity even when synthesized from same plant extract (ZE). This part has been incorporated in the revised manuscript.

References

8.Ratan, Z.A.; Haidere, M.F.; Nurunnabi, M.; Shahriar, S.M.; Ahammad, A.J.S.; Shim, Y.Y.; Reaney, M. J.T.; Cho, J.Y. Green Chemistry Synthesis of Silver Nanoparticles and Their Potential Anticancer Effects. Cancers. 2020,12,855

13.Mukherjee, S.; Chowdhury, D.; Kotcherlakota, R.; Patra, S.; B, V.; Bhadra, M.P.; Sreedhar, B.; Patra, C.R. Potential Theranostics Application of Bio-Synthesized Silver Nanoparticles (4-in-1 System). Theranostics. 2014,4,316-335.

14.Mukherjee, S.; Sushma, V.; Patra, S.;Barui, A.K.; Bhadra, M.P.; Sreedhar, B.; Patra, C.R. Green chemistry approach for the synthesis and stabilization of biocompatible gold nanoparticles and their potential applications in cancer therapy. Nanotechnology. 2012, 23, 455103.

15.Nethi, S. K.; Mukherjee, S.; Veeriah, V.; Barui, A. K.; Chatterjee, S.; Patra, C. R. Bioconjugated gold nanoparticles accelerate the growth of new blood vessels through redox signaling. Chem. Commun., 2014, 50, 14367-14370.

29.Kotcherlakota, R.; Nimushakavi, S.; Roy, A.; Yadavalli, H.C.; Mukherjee, S.; Haque, S.; Patra, C.R. Biosynthesized Gold Nanoparticles: In Vivo Study of Near-Infrared Fluorescence (NIR)-Based Bio-imaging and Cell Labeling Applications. ACS Biomater. Sci. Eng. 2019,5,5439-5452.

30.Mohamed, A.H.; Ahmed, F.A.; Ahmed, O.K. Hepatoprotective and Antioxidant Activity of Zinnia Elegans Leaves Ethanolic Extract. Int. J. Sci. Eng. Res. 2015, 6, 154-161.

31.Gomaa, A.A.R.; Samy, M.N.; Desoukey, S.Y.; Kamel, M. A comprehensive review of phytoconstituents and biological activities of genus Zinnia. J. Adv. Biomedical Pharm. Sci. 2019,2,29-37

52.Mukherjee, S. and C.R. Patra, Biologically synthesized metal nanoparticles: recent advancement and future perspectives in cancer theranostics. Future Sci. OA. 2017, 3, FSO203-FSO203.

Comment 2:

  1. i) The characterization of the silver NPs is not described, instead there is a reference to the characterization of gold NPs instead. I dont think this suffices for the purpose of this manuscript: the reader would really want to see the structure, shape, sizes etc. of the silcer NPs that are generated secifically (?) when the Zinnia extracts are used.

  1. ii) Why Zinnia? How has this been selected, was there a screening process? That, however, is provided nowhere.

iii) I agree with the authors that maybe the structure of the NPs formed with Zinnia extracts is similar and comparable to the ones generated with other plants. But – its simply not demonstrated how and if these NPs are different, have more beneficial properties that others, etc.

  1. iv) Isnt it possible that ANY plant extracts will contain sufficient reducing biomolecules that will resiult in the reduction of silver ions and precipitation of nanoparticles that have a mixed and probably very heterogeneous content? It is hinted in the manuscript to the differential reductive potential of plant extracts, leading to different sized NPs; but this may be a rather generic observation and all plant extracts are likely to contain reductive reagents (polyphenoles etc.).

Therefore, better characterization of this material in comparison to the previously published NPs, and other methods of their generation is an absolute MUST before this manuscript can be published. It almost makes no sense to even go through the results before this isnt properly done. There is no way other researchers could follow up on this without additional information.

Response 2:

  1. i) The biosynthesized silver nanoparticles (AgZE) are thoroughly characterized using several analytical tools (XRD, DLS, UV Visible, FTIR, SEM, TEM, and ICPOES) and the data is already presented in the main manuscript and supporting information. Furthermore, some characterizations (SEM, FTIR and ICPOES) are repeated and updated in the revised manuscript.

The above characterization techniques of the nanomaterials demonstrate the crystallinity, size, shape, surface charge of AgZE synthesized from Zinnia elegans. Cellular internalization of the fluorescent nanoparticles is observed by in vitroconfocal microscopy in normal (CHO) and cancer (U-87 and B16F10) cells and in vivo NIR based imaging in C57BL6/J mice model (melanoma). All data have been incorporated in the revised manuscript.

  1. ii) The Zinnia elegans plant extract has been selected since the ZE extract contains different phytochemicals, polyphenolic/alcoholic compounds and aldehydes/ketones that help in the reduction of Ag2+ (in AgNO3) to make silver nanoparticles (Ag0) [8,29,31]. Also, the ZE plant has several medicinal values such as hepatoprotective, antifungal, antihelmintic, antimalarial, anti-infective, phytoremediation, etc. [29-31]. Our hypothesis is that during synthesis, some of the bioactive molecules can attach on the surface of the nanomaterials that shows some biological activity. Also, ZE extract shows fluorescence in the NIR region as per published literature [29]. Our group already reported for biosynthesized gold nanoparticles (AuZE) obtained by the reduction of gold salts using extracts of Zinnia elegans, that exhibit fluorescence at both green (emission: 450 nm; excitation: 350 nm) and red region (emission: 720 nm; excitation: 450 nm) [29]. However, gold nanoparticles do not exhibit anticancer activity unless we use any chemotherapeutic drug.

On the other hand, anticancer activity of silver is well known. Considering the NIR-based bioimaging and medical values of ZE as well as anticancer activity of AgNP, the biosynthesized AgZE NPs could be useful for theranostic application with enhanced therapeutic efficacy and diagnostic applications in the near future.

We have also used other plant extract that does not show anticancer activity or NIR based imaging.

iii) In this present manuscript, shape of biosynthesized silver nanoparticles is spherical with tails produced using ZE extract. Olax scandens extract also produced similar shape and size (almost) silver nanoparticles exhibiting anticancer property against melanoma cancer model [13]. However, that does not show fluorescence imaging in the NIR region instead show normal fluorescence properties. Similar shape of nanoparticles (here AgZE) does not show similar therapeutic efficacy and NIR based bioimaging.

          As per the reviewer’s comment, the AgZE nanoparticles have unique beneficial property of exhibiting efficient anticancer activity as well as in vivo NIR based bioimaging (excitation: 710 nm; emission: 820 nm). Furthermore, the mechanism behind the anticancer property of AgZE is probably through p53 and caspase-8 mediated apoptotic pathway.

  1. iv) Yes, it is possible that different plant extracts contain sufficient reducing biomolecules that will result in the reduction of silver ions and precipitation of nanoparticles that have a mixed and probably very heterogeneous content. Here, in the current manuscript differential reductive potential of ZE extract leads to different sized (large and small size peak) AgZE nanoparticles observed by DLS analysis.

But with various plants (taking into consideration different locations, seasons and plant parts) different types of silver nanoparticles are formed with varying shape, size and activity due to presence of proteins, flavonoids (pigments), alkaloids, terpenoids and their corresponding concentration.

References

8.Ratan, Z.A.; Haidere, M.F.; Nurunnabi, M.; Shahriar, S.M.; Ahammad, A.J.S.; Shim, Y.Y.; Reaney, M. J.T.; Cho, J.Y. Green Chemistry Synthesis of Silver Nanoparticles and Their Potential Anticancer Effects. Cancers. 2020,12,855

29.Kotcherlakota, R.; Nimushakavi, S.; Roy, A.; Yadavalli, H.C.; Mukherjee, S.; Haque, S.; Patra, C.R. Biosynthesized Gold Nanoparticles: In Vivo Study of Near-Infrared Fluorescence (NIR)-Based Bio-imaging and Cell Labeling Applications. ACS Biomater. Sci. Eng. 2019,5,5439-5452.

30.Mohamed, A.H.; Ahmed, F.A.; Ahmed, O.K. Hepatoprotective and Antioxidant Activity of Zinnia Elegans Leaves Ethanolic Extract. Int. J. Sci. Eng. Res. 2015, 6, 154-161.

31.Gomaa, A.A.R.; Samy, M.N.; Desoukey, S.Y.; Kamel, M. A comprehensive review of phytoconstituents and biological activities of genus Zinnia. J. Adv. Biomedical Pharm. Sci. 2019,2,29-37

Comment 3: Otherwise, the NPs used here may of course have potential. Some of the experiments performed are conclusive, others less. For example, whats the point of performing hemolysis assays? How is this connected to the anti-cancer activities observed?

Response 3: We are thankful to reviewer’s critical evaluation. Hemolysis is an in vivo model where cell membrane damage causes discharge of hemoglobin along with other cellular components into the plasma from the erythrocytes.Several anticancer drugs that are commercially available show toxicity towards erythrocytes causing changes in their discoid shape and induces hemolytic anemia [44-45]. The AgZE exhibits anticancer effect, therefore to analyze the in vivo biocompatibility of the AgZE on the normal erythrocytes hemolysis assay is performed using the mice RBC.

Reference

44.Silva, M.C.; Madeira, V.M.; Almeida, L.M.; Custódio, J.B. Hemolysis of human erythrocytes induced by tamoxifen is related to disruption of membrane structure. Biochim. Biophys. Acta (BBA) - Biomembr. 2000, 1464, 49-61.

45.Haley, K.M., Russell, T.B.; Boshkov, L.; Leger, R.M.; Garratty, G.; Recht, M.; Nazemi, K.J. Fatal carboplatin-induced immune hemolytic anemia in a child with a brain tumor. J. Blood Med. 2014, 5, 55-58.

Comment 4:

  1. i) The optical properties of these nanoparticles are interesting, especially if it should be true that there are differences in cytotoxicity between "normal" (HEK-293, CHO) and various cancer cells. But, is there also a potential explanation for this?

ii)Are similar properties observed with other NPs generated in a similar way? r is this re ally something outstanding that is specific for the Zinnia-NPs? These kind of controls are missing, and would make the story more interesting and relevant.

iii) By the way, does/response curves are also missing, or an estaimate to EC50 values. iv) The concentrations used should cover a borader range - this is illustrated with HeLa cells, where there is no difference between 1and 10 ul of reagent. There should be more exponential (base 2) increasees in the acive concentrations used.

  1. v) And the concentrations should be measured more precisely: The quantification "ul" is not precise and insufficient for this type of activity assays.

Response 4: We thank the reviewer for the critical comment.

  1. i) The cell viability assay (Figure 2and Figure S4) of AgZE show selective cytotoxicity towards the cancer cell lines compared to normal cells lines that is further validated by cellular uptake study using ICPOES analysis (Figure S6).The study exhibits the more uptake of AgZE in cancer cells compared to normal cells.

As per reviewer’s comment, we have also performed the fluorescence study in vitro in normal (CHO) and cancer (U-87 and B16F10) cells using confocal microscopy. The study depicted AgZE exhibited fluorescence at the NIR range at LDS 751 channel using the confocal microscopy. Though red fluorescence is observed in all the three cells, the cancer cells appear to be sick and losing its morphology indicating it is undergoing apoptosis as compared to the normal cells.

  1. ii) The optical properties of AgZE are analyzed as ZE extract solution exhibit fluorescence as per earlier report [29]. Different excitation ranges are applied from 350 nm to 650 nm, which reveals AgZE exhibit fluorescence only at green channel (380 nm excitation and 450 nm emission) as shown in Figure S3. In our earlier report the ZE extract and AuZE exhibits fluorescence both at green (excitation-350 nm, emission-450 nm) and red (excitation-450 nm, emission-720 nm) channels [29]. But the AgZE shows fluorescence only at green channel in solution but NIR range bioimaging (both in vitro and in vivo), which may be due to the quenching of the fluorescence molecules (present in ZE extract) during the formation of silver nanoparticles.

                      There are reports of fluorescence properties exhibited by other biosynthesized nanoparticles or plant extracts. But their fluorescence does not fall into the NIR region. For example, earlier our group has synthesized silver nanoparticles using Olax scandens that exhibited fluorescence property in the red region of 605 nm (red region) [13]. The ZE extract particularly exhibited fluorescence in the NIR region. Our group for the first time reported for biosynthesized gold nanoparticles (AuZE) obtained by the reduction of gold salts using extracts of Zinnia elegans, that exhibit fluorescence at both green (emission: 450 nm; excitation: 350 nm) and red region (emission: 720 nm; excitation: 450 nm) [29]. This NIR based fluorescence is specific for ZE extract.

iii) The dose response curves for AgZE (as shown below) have been provided in the revised supporting information file in Figure S4 and S5.

  1. iv) The AgZE exhibited IC50 at 4.1μl corresponding to the 0.94 μg/ml (as per ICPOES results) in the U-87 cancer cell line, therefore we have not exceeded the dose beyond 10 ul. Therefore, the concentrations for AgZE are kept maximumupto10 ul.

  1. v) In order to maintain the numerical values in whole number we have used μl, however, the exact concentration of AgZE is calculated as per ICPOES results and provided in the corresponding manuscript.

References

13.Mukherjee, S.; Chowdhury, D.; Kotcherlakota, R.; Patra, S.; B, V.; Bhadra, M.P.; Sreedhar, B.; Patra, C.R. Potential Theranostics Application of Bio-Synthesized Silver Nanoparticles (4-in-1 System). Theranostics. 2014,4,316-335.

29.Kotcherlakota, R.; Nimushakavi, S.; Roy, A.; Yadavalli, H.C.; Mukherjee, S.; Haque, S.; Patra, C.R. Biosynthesized Gold Nanoparticles: In Vivo Study of Near-Infrared Fluorescence (NIR)-Based Bio-imaging and Cell Labeling Applications. ACS Biomater. Sci. Eng. 2019,5,5439-5452.

Comment 5: The cellular uptake assays are also poorly described, and the methods are soe what cryptic: "The samples are calculated as pulled data hence no standard deviation has been provided" (line395). This needs to be improved.

Response 5: As per the reviewer’s suggestion, the cellular uptake analysis now has been performed in triplicate (HEK and U-87) and provided in the revised supporting information as Figure S6.

Comment 6: Some other experimental approaches are convincing and informative, such as the scratch wound assays, the cell cycle/apoptosis analyses, and the ROS assays - pointing to rather generalized cytotoxicity, maybe. This specifically applies to the experiments deon with thre U-87 cell line; here the used controls and experimental conditions appear appropriate and sufficcient. Nevertheless, it may be an important issue (if the role of these NPsis further explored) to add more non-trnaformed, normal cells into these various assays formats, and validate there is a true difference in sensitivity between normal and cancer cells.

Response 6: We are thankful to the reviewer’s thoughtful suggestion. As per the suggestion, the cell viability assay using MTT (CHO and HEK-293), cell uptake assay (HEK-293) and confocal microscopy (CHO) are performed to determine the cytotoxicity of AgZE towards the normal and cancer cells. All the experiments MTT assay (Figure 2 and Figure S4), cellular uptake analysis (Figure S6) and confocal microscopy (Figure 8) indicates that AgZE is biocompatible with the normal cells. And among all cancer cells used, the AgZE is more cytotoxic to U-87 cell compared to other cancer cells hence it the further in vitro anticancer assays are performed only in the U-87 cancer cell line.

Comment 7:

  1. i) Concerning the Western blots, and the quantification: I cannot see and significant differences?
  2. ii) Why have these genes be selected to be investigated? (I can kind of understand caspase 8).

Response 7:We are thankful to the reviewer’s critical evaluation.

  1. i) As per the comment, the western blot analysis is performed to identify the key specific proteins that might be involved behind the anticancer activity of AgZE towards the U-87 cell line. The lanes of the western blot analysis are calculated as the ratio of test proteins values to loading control protein values (here GAPDH) as percentage of normalization, quantified using ImageJ analysis. Even if the lanes show same intensity when we divide each test protein with loading control the results are different.

% of Normalization: values of test protein/values of loading control protein

  1. ii) The upregulation and downregulation of proteins helps to give an overall idea through which signalling pathway a molecule is functioning. The U-87 cells treated with AgZE illustrates upregulation of apoptotic protein caspase-8 (first blot, third column) and tumor suppressor protein, p53 (second blot, third column) with the downregulation of STAT3 (third blot, third column) in comparison to untreated cells (first column of each blot) as seen from the blot images of Figure 7a. The STAT3 suppression can lead to the increase in the p53 since STAT3 binds to promoter region of the p53 in the DNA. The caspase-8 is an important upstream mediator in the death receptor mediated apoptosis. On the other hand, p53 and its family members are the upstream regulators of caspase-8 dependent apoptosis pathway. Therefore, caspase-8 is one of the mediators of the p-53 dependent apoptosis pathway [55]. We have demonstrated that the probable anticancer activity of AgZE is due to p53 dependent caspase-8 mediated apoptosis pathway. The increase in ROS by the AgZE might lead to the p53 upregulation and activation of the caspase-8 signaling pathway, leading to apoptosis.

References

55.Liu, J., Uematsu, H.; Tsuchida, N.; Ikeda, M.A. Essential role of caspase-8 in p53/p73-dependent apoptosis induced by etoposide in head and neck carcinoma cells. Mol. Cancer, 2011, 10, 95.

Comment 8:

  1. i) How does the growth of bllod vessels in CAM assay relate to the previous experiments done with cancer cell lines? There doesn’t appear to be a logical connection? (Leaky and defective bllod vessels in cancers?).
  2. ii) Plus, is there any significant difference? I am not convinced there is.

Response 8:We appreciate reviewer’s thoughtful evaluation.

  1. i) Tumor angiogenesis is one of the key factors for the progression of cancer through formation of new blood vessels from the pre-existing blood vessels. The chicken egg CAM assay model is a well-established study for evaluating anticancer activity [41, 42]. Here, we have just injected the nanoparticles to visualize the effect of blood vessel inhibition (supporting anticancer activity) by the nanoparticles. Therefore, the AgZE can prevent the formation of new blood vessels required for the growth and survival of the cancer cells and can lead to cell death.

  1. ii) Tumor angiogenesis is one of the key factors for the progression of cancer through formation of new blood vessels from the pre-existing blood vessels. The chicken egg-based CAM assay model is a well-established study for evaluating anticancer activity. Here, only the nanoparticles are incubated to visualize the effect of blood vessel inhibition by the nanoparticles for prevention of tumor angiogenesis [41, 42]. The CAM assay results as shown in Figure 8a indicates a decrease in the growth of blood vessels (w.r.t. length, size and junction) when treated with different concentrations of AgZE (2.5, 5 and 10 μl) compared to the untreated ones and analyzed by the Angioquant software. The 10 μl concentration also shows more inhibition with respect to other concentration due to increase in the amount of compound.

References

41.Vu, B.T.; Shahin, S.A.; Croissant, J.; Fatieiev, Y.; Matsumoto, K.; Doan, T.L.H.; Yik, T.; Simargi, S.; Conteras, A.; Ratliff, L.; Jimenez, C.M. Chick chorioallantoic membrane assay as an in vivo model to study the effect of nanoparticle-based anticancer drugs in ovarian cancer. Sci. Rep.2018, 8, 8524.

42.Merlos Rodrigo, M.A.; Casar, B.; Michalkova, H.; Jimenez Jimenez, A.M.; Heger, Z. ; Adam, V. Extending the Applicability of In Ovo and Ex Ovo Chicken Chorioallantoic Membrane Assays to Study Cytostatic Activity in Neuroblastoma Cells. Front. Oncol. 2021, 11, 707366.

.

Comment 9: The in vivo imaging then appears rather relevant and conclusive again; although the quality of the images can be improved. It wozkd all make much more sense, however, it mice with tumor xenografts woud have been used (for example, with the U-87 cell line) and there would have been significant incorpoation or enrichment of the NPs in comparison to any other tissues. That, however, has not been done.

Response 9: We thank the reviewer for constructive suggestion. As per reviewer suggestion for the proof of concept, a melanoma tumor model using B16F10 (mouse specific) cell line in the C57BL6/J mice have been generated after intraperitoneal injection of AgZE to exhibit more incorporation of the nanoparticle within the tumor as compared to the normal untreated tumor group.

Recently, we would to like to inform the reviewer that we have already initiated for the mice with tumor model for in vivobiodistribution, tumor regression studies, survivability and molecular mechanisms which will be a separate communication.

Comment 10: The paragraph on possible reasons for the cancer-specific action of the NPs is all speculation, none of this has been proved and demonstrated by any of the experiments shown in the manuscript. This section is not helping.

Response 10: The cancer-specific actions of nanoparticles are validated by several in vitro assays (MTT, cell cycle, apoptosis, ROS assays, western blot) and in vivo assay (CAM assay). We have also updated the corresponding sections in the revised manuscript.

Comment 11: English language use is sometimes a bit "funny", and requires moderate changes. There some odd typos in some sentences, like missing words and articles or prepositions etc., or "its" instead of" their". There are many of these, none really terrible but it does affect the overall impression. The discussion is largely a recapitulation of the results, and not really a discussion highlighting pros and cons of the study, or possible linke to existing literature by others. That can also be improved.

Response 11: The English language of the whole manuscript has been revised with the help of professional English speaker. As per the reviewer’s comment, we have taken care of all other issues in the revised manuscript. The pros and cons of the study have been provided in the preclinical section of the manuscript.

Finally, we are grateful to the Editor and reviewers for their critical comments, suggestions and advice that enormously helped us to improve the quality of the revised manuscript.

Hope the manuscript is now suitable for publication in ‘Cancers’

Round 2

Reviewer 1 Report

Unfortunately the authors don't replied to my concerns. In the previous revision, I asked them 2 principal modification. 

1) I think that the wound assay does not demonstrate that AgZE inhibits cell migration and, considering that the wound assay is unclear, it's important to confirm their result with another migration assay (you can ask to the journal a deadline extension). For me it is not understandable how the wound width after 7h AgZE treatment is larger that the wound width at 0 time of the same treatment. Again, by comparing the wound width at 0 and 7h for untreated cells, I can not conclude that AgZE treatment inhibits cell migration because the cells don't migrate when untreated. Furthermore, the authors write: "Also, at 7 h there are smaller numbers of viable U-87 cells due to the AgZE treatment (since it is cytotoxic) compared to untreated cells leading to larger wound width" and this assertion make impossible conclude that AgZE inhibits cell migration but, most probably, kills the cells that can not migrate.

2)The western blots in fig2 and their graphical representation don't agree. If I analyze your WB with ImageJ, I obtain the opposite result. For this, it would be better for the authors ameliorate the quality of WB (by repeating the experiment).

Author Response

Reviewer(s)’ Comments to Author:

Reviewer 1:

Overall comment: Unfortunately the authors don't replied to my concerns. In the previous revision, I asked them 2 principal modification.

Overall response: We are extremely sorry for the inconvenience caused to you as we could not perform the transwell based assay because of some genuine difficulties. Now we have performed the transwell based assay by borrowing the essential items needed for the experiment from another institute. Additionally, we have repeated the western blot experiments as per your suggestions and advice. Finally, we are grateful to you for giving us the opportunity to revise the manuscript for publication in the esteemed journal Cancers’. We hope that you will now recommend the revised manuscript suitable for publication in ‘Cancers’.

Comment 1:

I think that the wound assay does not demonstrate that AgZE inhibits cell migration and, considering that the wound assay is unclear, it's important to confirm their result with another migration assay (you can ask to the journal a deadline extension). For me it is not understandable how the wound width after 7h AgZE treatment is larger that the wound width at 0 time of the same treatment. Again, by comparing the wound width at 0 and 7h for untreated cells, I cannot conclude that AgZE treatment inhibits cell migration because the cells don't migrate when untreated. Furthermore, the authors write: "Also, at 7 h there are smaller numbers of viable U-87 cells due to the AgZE treatment (since it is cytotoxic) compared to untreated cells leading to larger wound width" and this assertion make impossible conclude that AgZE inhibits cell migration but, most probably, kills the cells that cannot migrate.

Response 1: We are thankful for reviewer’s constructive comment. As per the reviewer’s comment, we have performed the transwell based migration assay (in addition to wound scratching assay: Figure 3) and new data has been incorporated in the Supporting Information (Figure S7).

The AgZE exhibits antimigration effect towards U87 cells is further validated qualitatively using transwell based migration assay as per the published literature [34-35] and data are provided in Figure S7. The representative images of migrated U87 cells stained with crystal violet clearly observed for 14 h indicates that AgZE prevents migration of the U-87 cells as compared to the untreated control cells and cells treated with temozolomide corroborating with the results of the scratch assay (Figure 3). Temozolomide is used as positive control.

                       As per your concerns, we have repeated the wound scratch assay for control untreated and AgZE treated cells. The images (control and AgZE treated) for wound scratch assay in Figure 3 have been replaced with new images for the untreated and the AgZE treatment. The new results of scratch assay show less migration of U-87 cells when treated with AgZE compared to control untreated cells.

References

  1. Deng, G.; Zhou, F.; Wu, Z.; Zhang, F.; Niu, K.; Kang, Y.; Liu, X.; Wang, Q.; Wang, Y.; Wang, Q. Inhibition of cancer cell migration with CuS@ mSiO(2)-PEG nanoparticles by repressing MMP-2/MMP-9 expression. Int. J Nanomed. 2017, 13, 103-116.
  2. Kovács, D.; Igaz, N.; Marton, A.; Rónavári, A.; Bélteky, P.; Bodai, L.; Spengler, G.; Tiszlavicz, L.; Rázga, Z.; Hegyi, P.; Vizler, C.; Boros, I. M.; Kónya, Z.; Kiricsi, M. Core-shell nanoparticles suppress metastasis and modify the tumour-supportive activity of cancer-associated fibroblasts. J Nanobiotechnol. 2020, 18, 18.

Comment 2: The western blots in fig2 and their graphical representation don't agree. If I analyze your WB with ImageJ, I obtain the opposite result. For this, it would be better for the authors ameliorate the quality of WB (by repeating the experiment).

Response 2: We are thankful to reviewer’s thoughtful suggestions. As per your concern, the densitometric analysis of the western blot (already performed) have been examined blindly by two other scholars using ImageJ analysis and the result is replaced by the new analysis data. The densitometric analysis is updated in the revised main manuscript (Figure 7).

       However, as per your advice we have repeated the western blot experiments. The new results (provided below) are in similar pattern with the earlier western blot results. Therefore, we have not incorporated the new data in the revised manuscript

Figure 1. Western blot analysis of the U-87 cell lysates where the groups are untreated (first column), temozolomide: TEMO, positive control (second column) and AgZE (third column) with blots of a) caspase 8 (first blot), p53 (second blot), STAT3 (third blot) and GAPDH (fourth blot), b) graphical representation of quantification of western blots using Image J analysis. Upon treatment with AgZE, U-87 cell protein lysates shows upregulation of caspase-8 as well as p53 with slight upregulation of Stat3 compared to untreated samples. GAPDH is used as loading control.

Finally, we are grateful to the Editor and reviewers for their critical comments, suggestions and advice that enormously helped us to improve the quality of the revised manuscript.

We hope that you will now find the revised manuscript suitable for publication in Cancers’.

Reviewer 2 Report

The authors have taken this reviewers comments very seriously (and probably also those of other reviewers), and very significantly updated and expanded the manuscript. They have added a number of novel sections and experiments, to address the specific questions, and clarify most or all of the comments that were raised. 

There is not really much left to be done; this manuscript is now quite convincing even though (in my opinion) the process of generating NPs by reducing silver ions with plant extrcts is never going to be a highly controlled process... likely raising problems if these NPs or similar ones should ever be utilized in clinical settings (= on patients). But I also agree with the reviewers that this may not be super-critical at this experimental stage. As the authors state themselves: "Our hypothesis is that during 
synthesis, some of the bioactive molecules can attach on the surface of the nanomaterials that shows some biological activity. " Yes, this will probably always remain a hypothesis as such things will be very difficult to analyze. This issue is now also discussed in much more detail in the discussion part (lines 794 - 805), which is appreciated. Lines 923 - 930 then further pick up related aspects, which also add to the "big picture" idea. 

It is also helping that the authors are now citing their own work more extensively, in order to explain the basis of their NP production and the nature of these NPs. This is not exceeding the usual degree of self-citation and was necessary for clarity.  

Some methods have been added (such as FTIR), but luckily, they do not expand the volume of the manuscript very much. Also, additional cell lines (B16F10) have been utilized, which are described in brief; not adding too much unnecessary text in the revised version. 

Scratch wound/cell motility assays have been expanded, or added, which (in my opinion) are valuable experiments. It is only the most aggressive and advanced tumor cell lines that show high degrees of invasion or motility, and it is most valuable to specifically target these cells - whenever possible. 

I dont agree with the authors that mechanisms such as caspase-8 and/or p53-mediated apoptosis are per se "pathways"; they are rather control & execution mechanisms culminating in cell death - but this is secondary. 

Some of the experiments added now must have doubtlessly existed already, but were only now added to the manuscript. But this is perfectly okay, and again, adds to the credibility of the manuscript as a whole. 

The contrast and brightness of Fig. 8 could be improved. It is currently difficult to see significant differences there. 

The article is now more voluminous and lengthy; especially the discussion section is rather detailed and even a bit convoluted. This could be counteracted by some shortening of passages. 

There is still a mistake concerning silver ions: "Ag2+" is unlikely to be present, it will be Ag+ only

Author Response

Reviewer 2:

Overall comment: The authors have taken this reviewers comments very seriously (and probably also those of other reviewers), and very significantly updated and expanded the manuscript. They have added a number of novel sections and experiments, to address the specific questions, and clarify most or all of the comments that were raised. There is not really much left to be done; this manuscript is now quite convincing even though (in my opinion) the process of generating NPs by reducing silver ions with plant extrcts is never going to be a highly controlled process... likely raising problems if these NPs or similar ones should ever be utilized in clinical settings (= on patients). But I also agree with the reviewers that this may not be super-critical at this experimental stage. As the authors state themselves: "Our hypothesis is that during synthesis, some of the bioactive molecules can attach on the surface of the nanomaterials that shows some biological activity. " Yes, this will probably always remain a hypothesis as such things will be very difficult to analyze. This issue is now also discussed in much more detail in the discussion part (lines 794 - 805), which is appreciated. Lines 923 - 930 then further pick up related aspects, which also add to the "big picture" idea. It is also helping that the authors are now citing their own work more extensively, in order to explain the basis of their NP production and the nature of these NPs. This is not exceeding the usual degree of self-citation and was necessary for clarity. Some methods have been added (such as FTIR), but luckily, they do not expand the volume of the manuscript very much. Also, additional cell lines (B16F10) have been utilized, which are described in brief; not adding too much unnecessary text in the revised version. Scratch wound/cell motility assays have been expanded, or added, which (in my opinion) are valuable experiments. It is only the most aggressive and advanced tumor cell lines that show high degrees of invasion or motility, and it is most valuable to specifically target these cells - whenever possible.

I dont agree with the authors that mechanisms such as caspase-8 and/or p53-mediated apoptosis are per se "pathways"; they are rather control & execution mechanisms culminating in cell death - but this is secondary. Some of the experiments added now must have doubtlessly existed already, but were only

now added to the manuscript. But this is perfectly okay, and again, adds to the credibility

of the manuscript as a whole.

Overall response: We are thankful to you for your nice compliments for our revised manuscript. We are also grateful to you for giving us the opportunity to revise the manuscript before publication in the esteemed journal Cancers’. Based on your advice and instructions, we have revised the manuscript carefully and we fully agree with your comments, suggestion and advice. We hope that you will now recommend the revised manuscript suitable for publication in ‘Cancers’.

Comment 1:

  1. i) The contrast and brightness of Fig. 8 could be improved. It is currently difficult to see significant differences there.
  2. ii) The article is now more voluminous and lengthy; especially the discussion section is rather detailed and even a bit convoluted. This could be counteracted by some shortening of passages.

iii) There is still a mistake concerning silver ions: "Ag2+" is unlikely to be present, it will be Ag+ only.

Response 1: i) As per reviewer’s suggestion, we have updated the Figure 8 in the revised manuscript.

  1. ii) As per the reviewer’s suggestion we have updated the discussion section in the revised manuscript.

iii) We really regret for such typographical mistake. Now, we have taken care of this issue in the revised manuscript.

Finally, we are grateful to the Editor and reviewers for their critical comments, suggestions and advice that enormously helped us to improve the quality of the revised manuscript.

Hope the manuscript is now suitable for publication in ‘Cancers’
